# C/EBPβ enhances platinum resistance of ovarian cancer cells by reprogramming H3K79 methylation

Dan Liu[1], Xiao-Xue Zhang[1], Meng-Chen Li[1], Can-Hui Cao[1], Dong-Yi Wan[1], Bi-Xin Xi[1], Jia-Hong Tan[1], Ji Wang[1], Zong-Yuan Yang[1], Xin-Xia Feng[1], Fei Ye[1], Gang Chen[1], Peng Wu[1], Ling Xi[1], Hui Wang[1], Jian-Feng Zhou[1], Zuo-Hua Feng[2], Ding Ma[1] & Qing-Lei Gao[1]

Chemoresistance is a major unmet clinical obstacle in ovarian cancer treatment. Epigenetics plays a pivotal role in regulating the malignant phenotype, and has the potential in developing therapeutically valuable targets that improve the dismal outcome of this disease. Here we show that a series of transcription factors, including C/EBPβ, GCM1, and GATA1, could act as potential modulators of histone methylation in tumor cells. Of note, C/EBPβ, an independent prognostic factor for patients with ovarian cancer, mediates an important mechanism through which epigenetic enzyme modifies groups of functionally related genes in a context-dependent manner. By recruiting the methyltransferase DOT1L, C/EBPβ can maintain an open chromatin state by H3K79 methylation of multiple drug-resistance genes, thereby augmenting the chemoresistance of tumor cells. Therefore, we propose a new path against cancer epigenetics in which identifying and targeting the key regulators of epigenetics such as C/EBPβ may provide more precise therapeutic options in ovarian cancer.

[1] Cancer Biology Research Center (Key Laboratory of the Ministry of Education), Tongji Hospital, Tongji Medical College, Huazhong University of Science and Technology, Wuhan 430030, People's Republic of China. [2] Department of Biochemistry and Molecular Biology, Tongji Medical College, Huazhong University of Science and Technology, Wuhan 430030, People's Republic of China. These authors contributed equally: Dan Liu, Xiao-Xue Zhang. Correspondence and requests for materials should be addressed to Q.-L.G. (email: qingleigao@hotmail.com)

Tumor cell resistance to chemotherapy is a major cause of poor prognosis and high mortality in a broad array of human malignancies[1–4]. An unstable genome is considered to provide numerous opportunities for tumor evolution, including those for overcoming drug treatment[5,6]. Drug resistance is a multifactorial problem determined by many drug-resistance genes[7,8]. Therefore, focusing on only a single drug-resistance gene or pathway is thought to greatly oversimplify the actual conditions encountered in cancer and is unlikely to reveal an ideal therapeutic target[9]. In the emerging approach of precision medicine, a cocktail of drugs targeting multiple factors may be needed to overcome drug resistance in patients with cancer, and identifying the leading factors that target multiple drug-resistance genes may significantly improve therapeutic efficacy[10].

Epigenetic reprogramming of gene expression patterns via modification of histones and/or DNA is an important mechanism underlying the simultaneous modulation of numerous genes, and plays a pivotal role in regulating the malignant phenotype[11–13]. Previous studies in epigenetics mainly focused on epigenetic enzymes that may also act as master regulators. However, a limited number of, or even only one, enzyme(s) catalyze a specific epigenetic site[14]. Thus, targeting these enzymes, controlling the relevant epigenetic sites in the whole genome, is often non-precise. The mechanisms, involving enzyme cofactors, that underlie the context-dependent and/or sequence-specific regulation of epigenetics are a key issue in the field, and are just beginning to be elucidated[15,16]. Furthermore, the mechanisms underlying the reprogramming of multiple drug-resistance genes are unknown.

Epigenetic alterations are associated with drug resistance in various cancers including ovarian cancer[9,17]. Ovarian cancer is the leading cause of death from gynecological malignancies[18], and the current standard treatment is platinum-based chemotherapy following surgical debulking. High-grade serous ovarian cancer (HG-SOC), the most prevalent and highly malignant type of ovarian cancer, is a typical solid tumor with a highly aberrant genome[5,11,19]. For decades, intrinsic and inevitably acquired resistance to chemotherapy in the vast majority of patients has presented a major barrier for the successful treatment of ovarian cancer[1]. In this study, we performed a genome-scale interrogation of histone methylation profiles of purified primary ovarian cancer cells. Our findings indicate that a set of functionally related genes involved in epigenetic reprogramming are controlled by specific transcription factors (TFs). By recruiting the methyltransferase DOT1L, CCAAT/enhancer-binding protein β (C/EBPβ, also known as CEBPB) could maintain an open chromatin state (H3K79 methylation) at multiple drug-resistance genes, thereby augmenting chemoresistance of tumor cells. Thus, the activities of TFs such as C/EBPβ mediate important mechanisms through which epigenetic enzymes modify chromatin in a site-specific manner. Our results herein suggest that in addition to the epigenetic enzymes themselves, cofactors may be alternative options for more precise targeted therapy.

## Results

**Specific TFs correlate to epigenetic reprogramming in HG-SOC.** Increasing evidence indicates that a significant proportion of HG-SOC originates in the fimbriae of fallopian tube[20–22]. Tissue cells from patients with HG-SOC (Supplementary Table 1) and the fimbriae of fallopian tubes from age-matched patients with benign diseases (hysteromyoma and adenomyosis; see also the Methods section) were first sorted using antibodies targeting epithelial cell adhesion molecule (EpCAM)[23]. The purity of isolated cells (>90%) was verified using lineage-specific paired-box gene 8 (PAX8)[24] and the epithelium-specific marker EpCAM (Supplementary Figure 1a). Tissue type-specific expression profiles of the purified cells, which were analyzed by transcriptome sequencing (RNA-sequencing (RNA-seq)), also showed high cell purities (Supplementary Figure 1b). Mutations in *TP53*, which are

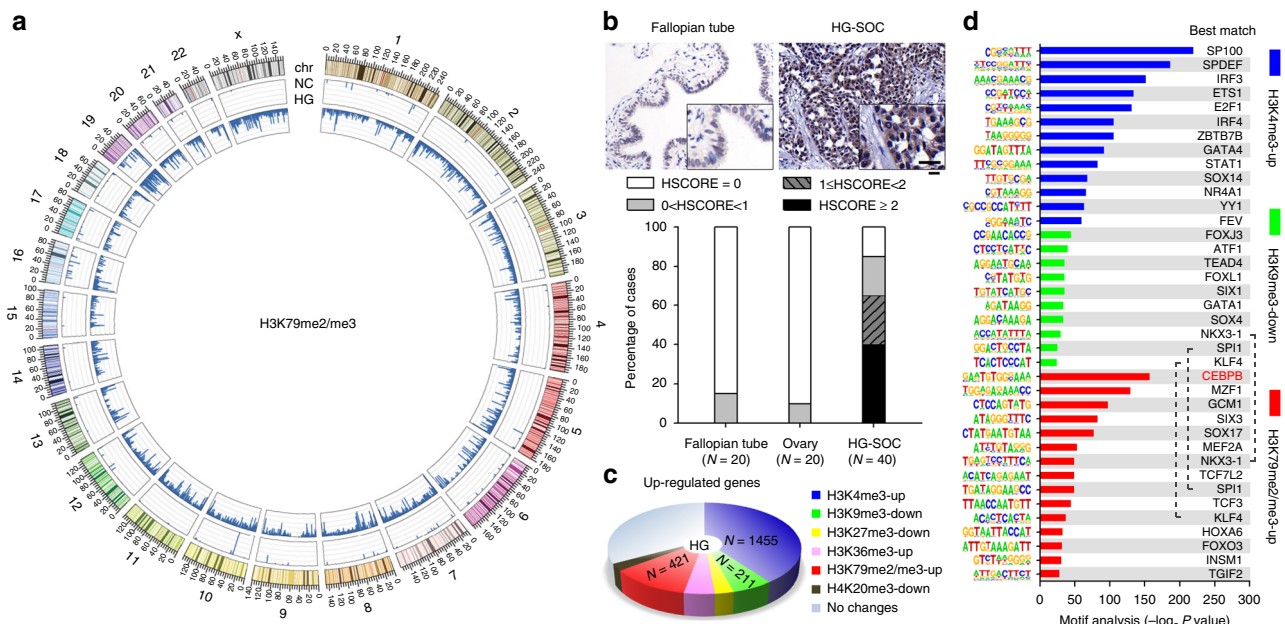

**Fig. 1** Specific TFs correlate with epigenetic reprogramming in HG-SOC. Magnetic separation of epithelial cells and preparation of pooled samples from 20 HG-SOC and 20 normal fallopian tube samples were performed as described in the Methods. **a** Global profiles for H3K79me2/me3 ChIP-seq in purified and pooled samples. The histogram axis scale represents read density per million sequenced reads and the outer DNA numbering is given in millions of bases. **b** IHC analysis of H3K79me2/me3 in human HG-SOC, borderline serous ovarian tumors, and normal control samples. Bar, 25 μm. **c** Pie diagrams showing the numbers of upregulated genes coupled with corresponding changes in the indicated histone methylation sites (HG-SOC vs. NC groups). **d** Summary of significantly enriched motifs (HG-SOC vs. NC groups) and the best matching TFs generated by de novo motif analysis

found in over 95% of cases with HG-SOC[25,26], were confirmed by whole-exon sequencing of each HG-SOC specimen (Supplementary Table 1). Compared to those found in normal fallopian tubes, differentially expressed genes in HG-SOC samples also showed a high enrichment in TP53-associated signaling pathways (Supplementary Figure 2).

To explore epigenetic reprogramming of HG-SOC in our patient cohort, we performed genome-scale analysis of the methylation patterns at six sites located within histones: trimethylated Lys4 (H3K4me3), trimethylated Lys36 (H3K36me3), and di- or trimethylated Lys79 (H3K79me2/me3) on histone H3 correlated with active transcription; H3K9me3, H3K27me3, and H4K20me3 correlated with transcriptional repression[27,28]. Chromatin immunoprecipitation (ChIP) coupled with high-throughput sequencing (ChIP-seq) revealed that HG-SOC contained elevated H3K4 and H3K79 methylation in many genes and decreased H3K9 methylation in some genes compared to those found in normal fallopian tubes (Supplementary Figure 3). Specifically, H3K79 exhibited the most dramatic and broadest change of any of the examined methylation sites in HG-SOC (Fig. 1a). Immunohistochemical (IHC) analysis also showed that global H3K79 methylation was low in the epithelium of normal fallopian tube and ovary, but was significantly increased in HG-SOC (Fig. 1b). Meanwhile, RNA-seq analysis of purified cells from fallopian tube and HG-SOC samples indicated that many upregulated genes in HG-SOC were associated with higher H3K4, higher H3K79, and lower H3K9 methylation (Fig. 1c), while many of the downregulated genes were associated with higher H3K9 methylation (Supplementary Figure 3d). Taken together, these results suggest that increased H3K79 methylation may play an important role in regulating gene expression in HG-SOC.

To identify key regulators of histone methylation in ovarian cancer, we investigated whether differentially methylated chromatin was significantly enriched in specific sequence motifs indicative of binding sites of TFs. Hypergeometric Optimization of Motif EnRichment (HOMER) de novo Motif analysis identified 38 enriched motifs corresponding to 35 TFs (Fig. 1d; $P < 1 \times 10^{-11}$). Among these TFs, CEBPB, which was upregulated in ovarian cancer (Supplementary Figure 4a), was most significantly correlated with increased H3K79 methylation (Fig. 1d; $P = 1 \times 10^{-86}$). Further bioinformatics analysis using The Cancer Genome Atlas (TCGA) dataset[25] predicted the downstream genes of these 35 TFs. CEBPB was shown to not only modulate the expression of many genes (Supplementary Figure 4b), but also regulate the expression of over one-third of H3K79-associated upregulated genes in HG-SOC (Supplementary Figure 4c). Of the other identified TFs, some including GATA4, IRF3, and SP100 were correlated with H3K4me3; GATA1 and others were correlated with H3K9me3; and TFs such as GCM1 were correlated with H3K79me2/me3 (Supplementary Figure 4d and 4e). These results indicate that specific TFs are correlated with histone methylation at the levels of both genome (sequence motif) and regulation of gene expression. To further understand the epigenetic reprogramming of HG-SOC, we investigated protein interactions among epigenetically altered genes and known important biological factors in this disease. Genes with increased H3K4 methylation showed high enrichment in those groups of genes that interact with MYC, TP53, BRCA1, RB1, and CCNE1 (Supplementary Figure 5). However, related genes (e.g., *CDK1*, *CDK2*, *SMARCB1*, *SMARCA4*) were excluded from the present study because they are either well-known genes in cancer or did not have significantly altered messenger RNA (mRNA) levels.

**C/EBPβ correlates with poor patient prognosis**. Since a regulatory factor may not actually play a significant role in determining tumor phenotype, we next investigated the relationships between these TFs and prognosis, one of the most important

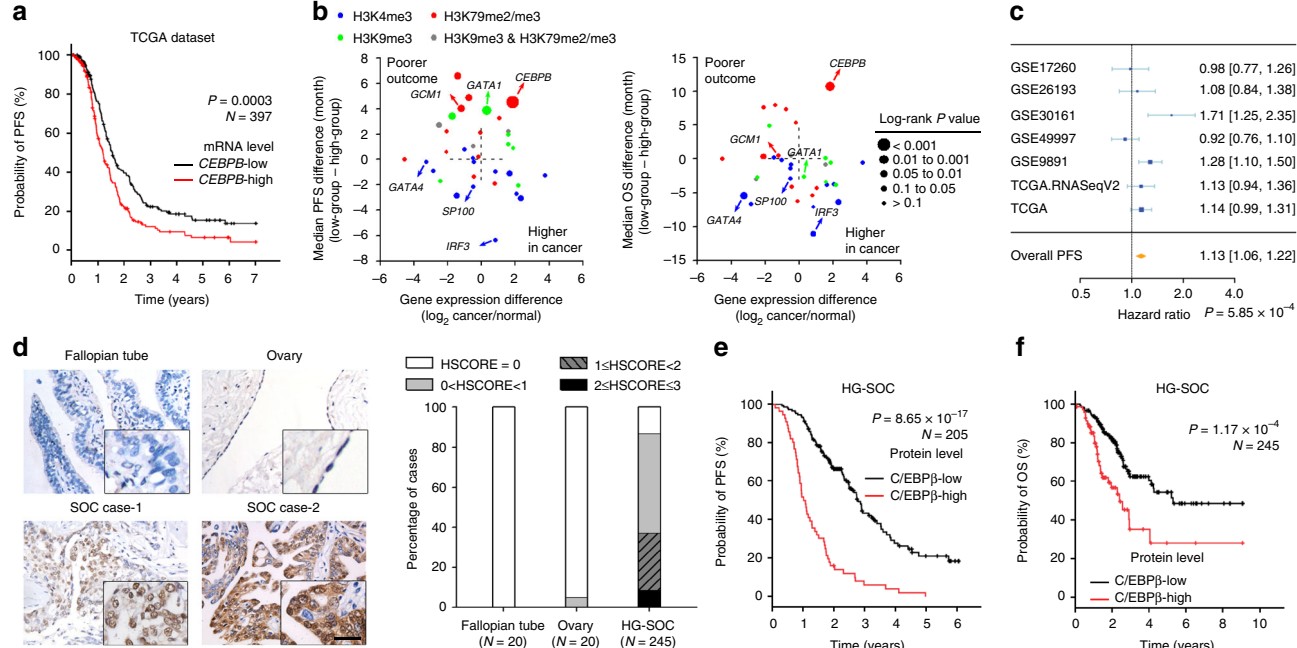

**Fig. 2** Evaluation of potential chromatin-modifying TFs. **a** Analysis of PFS using the TCGA dataset. Samples were divided into low- and high-expression groups based on the median expression of each gene. A survival curve of *CEBPB* is representatively shown. **b** Analysis of patient outcomes (differences in median survival times) vs. differences in gene expression. Left, progress-free survival; right, overall survival. **c** Gene expression meta-analysis of *CEBPB* in ovarian cancer. Horizontal lines represent 95% CI; diamonds represent summary estimates with the corresponding 95% CI. **d** IHC analysis of C/EBPβ protein in HG-SOC and normal control samples. Bar, 25 μm. **e, f** Kaplan–Meier analysis of PFS (**e**) and OS (**f**) in patients with ovarian cancer classified based on tumor C/EBPβ protein levels. PFS progress-free survival, OS overall survival

determinants of tumor malignancy. First, a preliminary analysis was performed using the TCGA dataset. Samples were divided into low- and high-expression groups based on the median expression of each gene. We found that higher *CEBPB* expression was correlated with shorter progress-free survival (PFS; Fig. 2a)

and shorter overall survival (OS; Supplementary Figure 6a) in patients with HG-SOC. Additionally, *IRF3* and *SP100* mRNA levels were not significantly associated with patient outcome, while the expression of *GCM1*, which was correlated with poorer PFS, was decreased in HG-SOC (Fig. 2b). In addition, *GATA4*

**Table 1 Multi-factor analysis of the prognosis in 245 patients with HG-SOC using the Cox regression**

| Factor | OS | | | PFS | | |
|---|---|---|---|---|---|---|
| | Relative risk | 95% CI | *P* | Relative risk | 95% CI | *P* |
| C/EBPβ expression (low vs. high) | 2.385 | 1.499–3.795 | 0.0002 | 4.201 | 2.928–6.028 | $6.72 \times 10^{-15}$ |
| Residual disease (R0 vs. R1) | 3.516 | 2.203–5.610 | $1.34 \times 10^{-6}$ | 2.176 | 1.523–3.109 | $1.93 \times 10^{-5}$ |
| Age at diagnosis (≤55 vs. >55 years) | – | – | 0.359 | – | – | 0.314 |
| FIGO stage (II vs. III, IV) | – | – | 0.053 | – | – | 0.821 |
| Ascites (no vs. yes) | – | – | 0.428 | – | – | 0.581 |

FIGO International Federation of Gynecology and Obstetrics, R0 no gross residual, R1 any residual

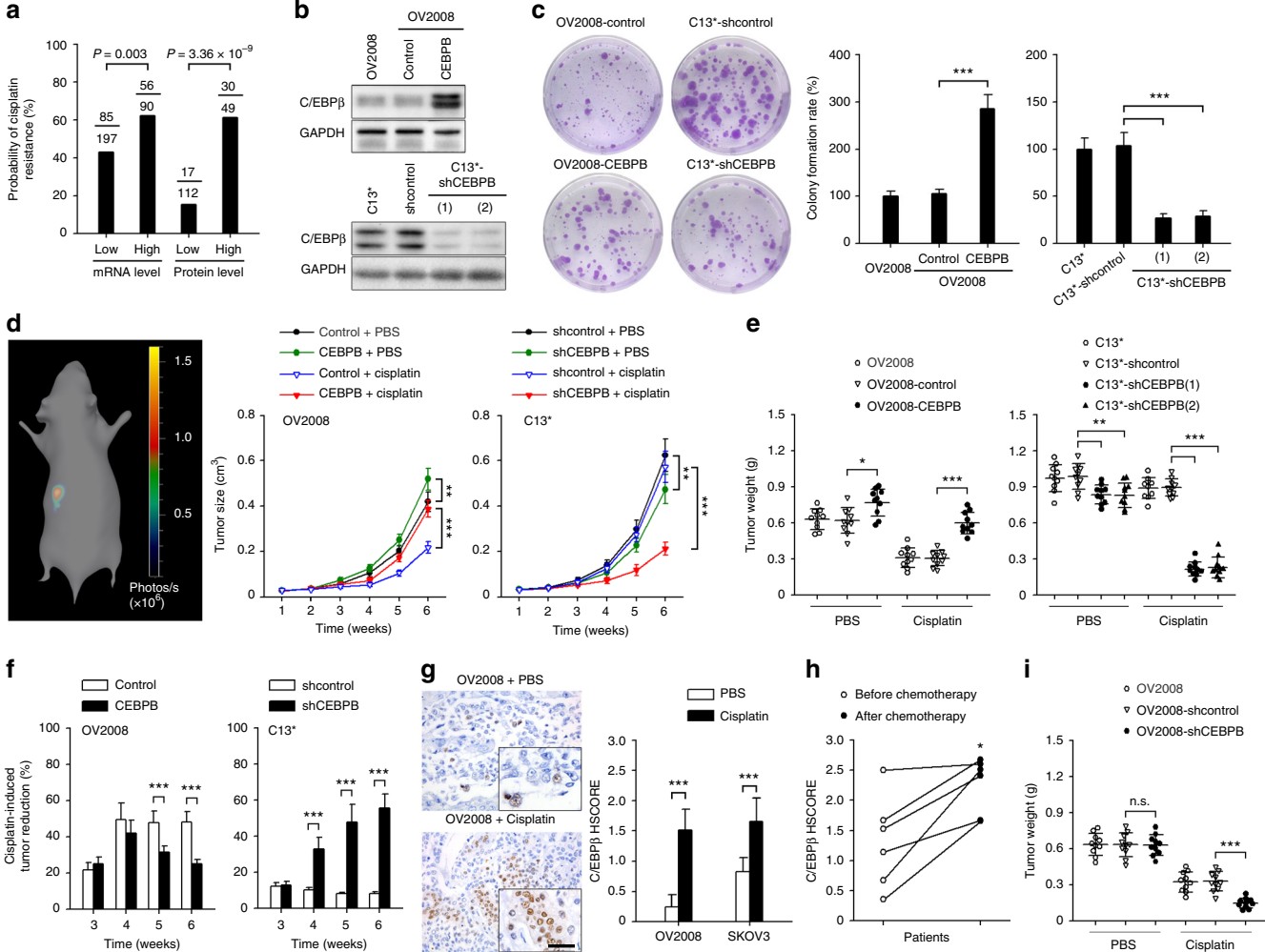

**Fig. 3** C/EBPβ promotes cisplatin resistance in ovarian cancer. **a** The probability of cisplatin resistance in patients with ovarian cancer in the CEBPB-low and CEBPB-high groups (Pearson's chi-squared test). mRNA and protein levels were analyzed using the TCGA dataset and IHC, respectively. **b** C/EBPβ expression was detected by western blotting. **c** Colony formation of the indicated cells after exposure to cisplatin (50 μM) for 12 h. **d**–**g** One week after orthotopic inoculation with the indicated cells, mice were treated with cisplatin (5 mg/kg) or vehicle (PBS) intraperitoneally every 4 days (n = 10 per group). Tumor size was monitored by three-dimensional reconstruction of in vivo bioluminescence images (**d**). Six weeks after tumor inoculation, tumors were excised and weighed (**e**). The cisplatin-induced tumor reduction rate was calculated as described in the Methods using the following formula: $(1 - V_{cisplatin}/V_{PBS}) \times 100\%$ (**f**). IHC analysis of C/EBPβ protein levels in xenograft tumor sections collected from mice treated with or without cisplatin. Bar, 25 μm (**g**). **h** IHC analysis of C/EBPβ protein in paired tissue sections collected from initial operation and reoperation performed for management of tumor recurrence in the same patient. **i** Tumor weights from mice treated as described in **d**–**g**. Six weeks after tumor inoculation, tumors were excised and weighed. Uncropped images of blots are shown in Supplementary Figure 25. *$P < 0.05$; **$P < 0.01$; ***$P < 0.001$; n.s. not significant

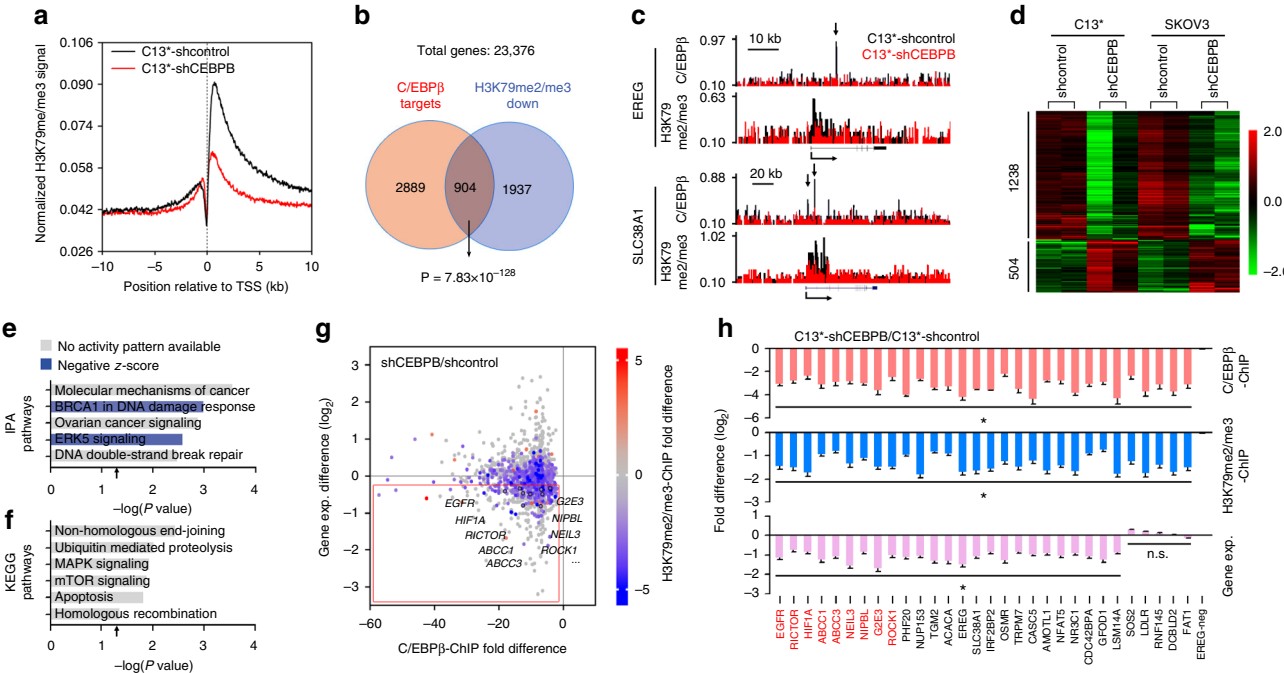

**Fig. 4** C/EBPβ modulates H3K79 methylation to reprogram gene expression. **a** Meta-analysis of the averaged H3K79me2/me3 ChIP-seq signals of genes across ±10 kb genomic regions flanking the TSS. **b** Venn diagrams showing the overlap of genes with decreased H3K79me2/me3 (C13*-shCEBPB vs. C13*-shcontrol) and C/EBPβ-targeted genes (Pearson's chi-squared test). **c** Normalized C/EBPβ and H3K79me2/me3 ChIP-seq signals for the representative genes (*EREG* and *SLC38A1*). **d** Supervised clustering of genes that were consistently differentially expressed (P < 0.05) upon C/EBPβ knockdown in C13* and SKOV3 cells. **e**, **f** IPA of the differentially expressed genes (**e**) and KEGG pathway analysis of downregulated genes (**f**) upon C/EBPβ knockdown in C13* and SKOV3 cells. Arrows indicate P = 0.05. Negative z-scores (blue) represent the suppressed state of the function. **g** Scatter plot showing genome-wide changes in the values of C/EBPβ binding (x-axis), gene expression (y-axis), and averaged H3K79me2/me3 (colored) in C13*-shCEBPB vs. C13*-shcontrol. Positive and negative values reflect increases and decreases, respectively. **h** Analysis of C/EBPβ ChIP-qPCR (pink), H3K79me2/me3 ChIP-qPCR (blue), and RT-qPCR (purple) in C13* cells. Gene names in red indicate documented cisplatin-resistance genes in ovarian cancer. *P < 0.05; n.s. not significant

and *GATA1* showed associations with either OS or PFS, but not both (Fig. 2b). Moreover, gene expression meta-analysis[29] using 4411 patients from 30 studies further demonstrated that *CEBPB* expression was a risk factor for the survival of patients with ovarian cancer (Fig. 2c and Supplementary Figure 6b). Consistent with the above results, IHC analysis of 277 human epithelial ovarian cancer specimens, including 245 specimens of HG-SOC (Supplementary Table 2), confirmed that the protein levels of C/EBPβ, which were negligible in normal fallopian tubes and ovarian epithelium, were significantly increased in ovarian cancer (Fig. 2d and Supplementary Figure 7). Classifying HSCORE <1.5 and HSCORE ≥1.5 as low and high protein levels respectively, we found no bias between low and high C/EBPβ protein groups with known major factors, specifically age, stage, ascites, and the amount of residual disease (Supplementary Table 2). However, we found that a high level of C/EBPβ protein was strongly associated with both poorer PFS (Fig. 2e) and poorer OS (Fig. 2f) in our patient cohort.

Univariate Cox regression analysis including C/EBPβ expression, residual disease, age, International Federation of Gynecology and Obstetrics (FIGO) stage, and ascites showed that C/EBPβ expression (low vs. high) and residual disease (R0 (no gross residual) vs. R1 (any residual)) were significantly associated with OS and PFS, and that age at diagnosis (≤55 vs. >55 years of age) and FIGO stage (II vs. III and IV) were significantly associated with OS (Supplementary Table 3). In addition, multivariate analysis demonstrated that C/EBPβ protein level was an independent prognostic factor for OS and PFS in patients with HG-SOC (Table 1). These findings demonstrate that C/EBPβ is correlated with both H3K79 methylation and patient survival.

Although we cannot exclude the possibility that other factors also contribute to various malignant phenotypes through modulation of histone methylation, we focused on C/EBPβ for the purpose of this study.

**C/EBPβ promotes cisplatin resistance.** Our analysis of human ovarian tumors indicated a higher probability of platinum resistance in patients exhibiting high levels of *CEBPB* mRNA (TCGA dataset) or C/EBPβ protein (IHC analysis) (Fig. 3a). Consistent with this determination, we found that endogenous C/EBPβ levels were associated with cisplatin resistance in ovarian cancer cell lines (Supplementary Figure 8). To further confirm that C/EBPβ promotes cisplatin resistance in ovarian cancer, C/EBPβ was overexpressed in OV2008, which is a cisplatin-sensitive cell line with negligible C/EBPβ expression, and C/EBPβ was knocked down in C13* cells, which are cisplatin-resistant derivatives of OV2008 cells that exhibit strong C/EBPβ expression (Fig. 3b). We found that C/EBPβ overexpression significantly enhanced cell viability and higher colony formation rates as well as markedly decreased cell apoptosis in response to cisplatin treatment (Fig. 3c and Supplementary Figure 9). Consistently, knockdown of C/EBPβ in C13* cells resulted in decreased cisplatin resistance in vitro (Fig. 3c and Supplementary Figure 9). Upon cisplatin treatment, tumor size (Fig. 3d) and final tumor weight (Fig. 3e) in orthotopically implanted tumors in mice were significantly increased when C/EBPβ was overexpressed in OV2008 cells and decreased when C/EBPβ was knocked down in C13* cells. However, it was also shown that C/EBPβ alone promoted tumor growth in vivo (Fig. 3d, e). To exclude the change in proliferation rate because of changes to C/EBPβ protein levels, a

cisplatin-induced tumor reduction rate was calculated based on the tumor volume in a phosphate-buffered saline (PBS)-treated group at each time point. Our results consistently showed that the rate of tumor reduction was decreased when C/EBPβ was over-expressed in OV2008 cells and increased when C/EBPβ was knocked down in C13[*] cells (Fig. 3f). The same effect was also observed in SKOV3 and Caov3 ovarian cancer cells (Supplementary Figures 9 and 10). These results demonstrate that C/EBPβ promotes cisplatin resistance in ovarian cancer.

The above data showed that C/EBPβ knockdown dramatically increased cisplatin sensitivity in vivo, while minor differences in cisplatin sensitivity were observed between C/EBPβ overexpression and control groups. We found that C/EBPβ expression was significantly increased after cisplatin treatment in vitro (Supplementary Figure 10e) and in vivo (Fig. 3g). Elevated C/EBPβ expression after platinum-based chemotherapy was also observed in paired ovarian cancer specimens (Fig. 3h), a finding which was consistent with higher cell viability in C/EBPβ-expressing cells. Thus, under the pressure of cisplatin, a tumor may become more resistant over time by selecting C/EBPβ-expressing cells. Additionally, in OV2008 control cells, further reductions in tumor weight upon cisplatin treatment were observed when C/EBPβ expression was blocked by small hairpin RNA (shRNA; Fig. 3i and Supplementary Figure 10f). These results suggest that the shift in C/EBPβ expression upon cisplatin treatment may play an important role in acquired resistance to platinum-based agents in ovarian cancer. In addition, knockdown of C/EBPβ increased sensitivity to carboplatin, which has an antitumor mechanism similar to cisplatin (Supplementary Figure 11a). Regarding other commonly used chemotherapy drugs in ovarian cancer, our preliminary data showed that C/EBPβ levels have no statistically significant impact on paclitaxel, doxorubicin, and topotecan sensitivity (Supplementary Figure 11b–d).

It was reported that post-transcriptional regulation was a key mechanism for the regulation of C/EBPβ protein, and that mRNA levels would not necessarily be regulated[30]. In this context, although we found that the level of C/EBPβ protein was remarkably higher in C13[*] cells compared to that found in its parent OV2008 cells, there was no difference in CEBPB mRNA levels between the two cell lines (Supplementary Figure 12a). Similarly, there were no significant changes in CEBPB mRNA levels after cisplatin treatment in vitro (Supplementary Figure 12b) and in vivo (Supplementary Figure 12c). Using an independent dataset of ovarian tumors[26], CEBPB mRNA levels were also shown to be slightly but not significantly elevated in recurrent diseases compared to that found in primary tumors (Supplementary Figure 12d). These results suggest that changes to C/EBPβ after platinum exposure may be primarily because of post-transcriptional mechanisms. However, CEBPB mRNA was significantly increased in ovarian cancer (Supplementary Figure 12e), and there was a positive correlation between C/EBPβ mRNA and protein levels among cell lines (Supplementary Figure 12a) and clinical specimens (Supplementary Figure 12f), indicating the involvement of transcriptional regulation of C/EBPβ. Therefore, there may be different mechanisms for C/EBPβ upregulation in tumorigenesis and after chemotherapy, the investigation of which is beyond the scope of the present study.

**C/EBPβ reprograms H3K79 methylation**. We next sought to determine whether C/EBPβ modulated H3K79 methylation. ChIP-seq analysis indicated that knockdown of C/EBPβ in C13[*] cells resulted in a general decrease in H3K79 methylation in proximal regions downstream of the transcription start sites (TSSs) (Fig. 4a and Supplementary Data 1). Moreover, we identified 6849 C/EBPβ peaks distributed in 3793 protein-coding genes

in C13[*] cells (Supplementary Data 2). These C/EBPβ-targeted genes were significantly correlated with decreased H3K79 methylation in C/EBPβ-knockdown cells (Fig. 4b), a finding also represented by a genome-browser view of ChIP-seq signals where C/EBPβ binding sites resided within H3K79 methylation peaks (Fig. 4c and Supplementary Figure 13a). Similar results were obtained in SKOV3 cells (Supplementary Figure 13b and Supplementary Data 1 and 2).

On the other hand, a recent finding showing that the H3K79 methylation state prevents H3K9 methylation suggests that increased H3K79 methylation and decreased H3K9 methylation, both of which are associated with transcriptional activation, may coordinate to regulate gene expression[31]. Consistent with this finding, our correlation analysis using total genes (Supplementary Figure 13c) and upregulated genes (Supplementary Figure 13d) of sequencing data from patients with HG-SOC and controls with normal fallopian tubes showed that increased H3K79 methylation was associated with decreased H3K9 methylation in HG-SOC. To determine the role of C/EBPβ in regulating the two methylation states, H3K9 methylation ChIP-seq was also performed. We found that knockdown of C/EBPβ in C13[*] cells enhanced H3K9 methylation across entire genes (Supplementary Figure 13e and Supplementary Data 3). However, although some C/EBPβ-targeted genes exhibited increased H3K9 methylation in C/EBPβ-knockdown cells ($P = 3.35 \times 10^{-8}$, chi-squared tests), the C/EBPβ-associated H3K79me2/me3-downregulated genes and the C/EBPβ-associated H3K9me3-upregulated genes were independent of each other ($P =$ not significant, chi-squared tests; Supplementary Figure 13f). Similar data were obtained in SKOV3 cells (Supplementary Figure 13g and S13h and Supplementary Data 1–3). These results indicate distinct molecular mechanisms between C/EBPβ-induced H3K79 methylation changes and C/EBPβ-induced H3K9 methylation changes. Because a C/EBPβ-like motif was originally associated with H3K79 methylation and a greater regulatory effect was observed on H3K79 methylation, we then focused on the regulation of C/EBPβ on H3K79 methylation.

Because H3K79 methylation promotes gene expression[27], we performed RNA-seq analysis to identify C/EBPβ-regulated genes genome wide. We found that 1238 genes were consistently downregulated and 504 genes were consistently upregulated upon C/EBPβ knockdown in two ovarian cancer cell lines (Fig. 4d and Supplementary Data 4). Ingenuity pathway analysis (IPA) of the differentially expressed genes revealed high enrichment in cancer-associated pathways and predicted suppression of DNA damage repair (DDR) and extracellular signal-regulated kinase 5 (ERK5) signaling (Fig. 4e, Supplementary Figure 14, and Supplementary Data 5), both of which are critical for cisplatin resistance[1]. Our analysis of the downregulated genes also showed significant associations with DDR-related processes, such as ubiquitin-mediated proteolysis, nonhomologous end-joining, and homologous recombination[32,33] as well as survival-related signals (i.e., mitogen-activated protein kinase (MAPK), mammalian target of rapamycin (mTOR) signaling, and apoptosis) (Fig. 4f). In contrast, we found that upregulated genes were not enriched in any tumor-associated pathway (Supplementary Data 6). These findings indicate that C/EBPβ specifically activates oncogene expression.

Next, we performed a comprehensive investigation of histone methylation and gene expression in C13[*]-shCEBPB vs. C13[*]-shcontrol cells. We identified hundreds of C/EBPβ-targeted genes with decreased H3K79 methylation and decreased gene expression (Fig. 4g), of which 9 (indicated in Fig. 4g, h) are known cisplatin-resistance genes in ovarian cancer and another 21 genes are functionally correlated with cisplatin resistance (Supplementary Figure 15). ChIP-qPCR analysis confirmed the decreased C/

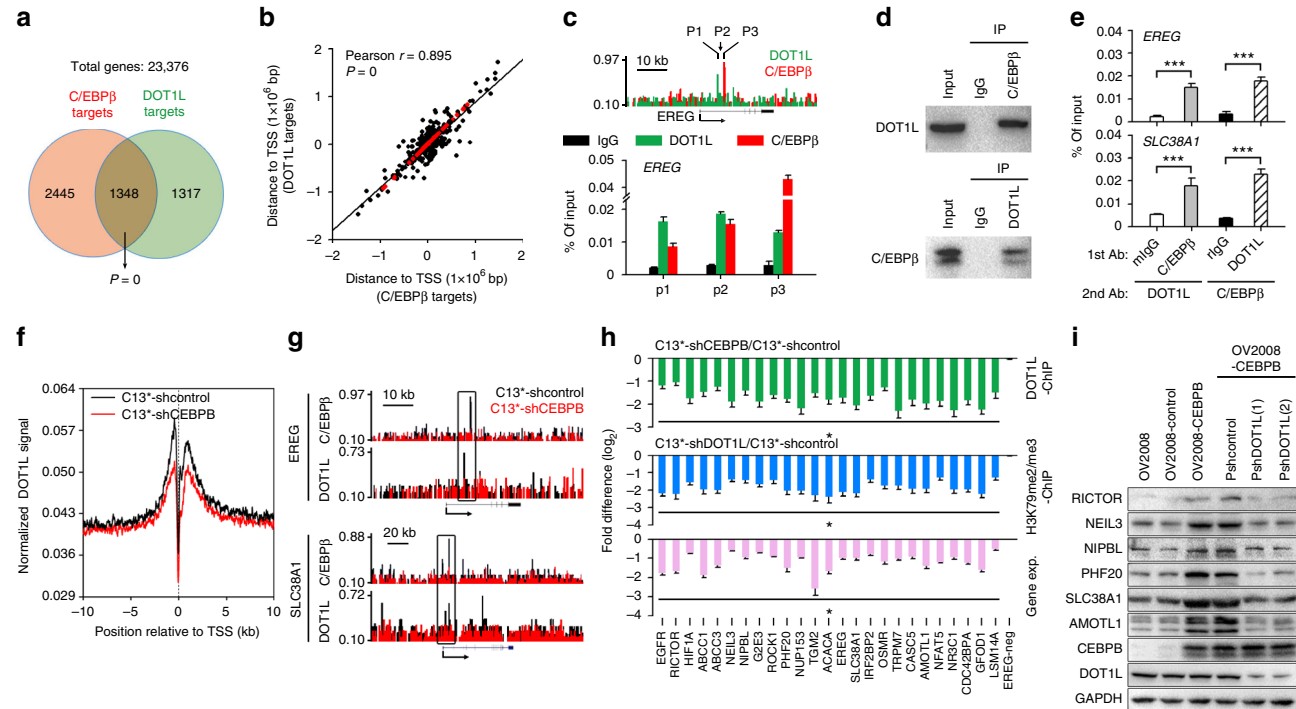

**Fig. 5** C/EBPβ recruits the methyltransferase DOT1L to target genes that methylate H3K79. **a** Venn diagrams showing the overlap of C/EBPβ- and DOT1L-targeted genes (chi-squared test). **b** Correlation analysis of C/EBPβ binding sites and DOT1L binding sites. Red dots represent genes with two peak centers located less than 2500 bp apart. **c** Representative results of C/EBPβ and DOT1L ChIP-qPCR. Primers were chosen according to the ChIP-seq results (upper panel). **d** C/EBPβ interacts with DOT1L. Lysates from C13* cells were immunoprecipitated (IP) with a mouse anti-C/EBPβ antibody and analyzed by western blotting using a rabbit anti-DOT1L antibody (upper panel) or the reciprocal (lower panel). **e** ChIP-reChIP experiments with anti-C/EBPβ and anti-DOT1L antibodies. Mouse IgG (mIgG) and rabbit IgG (rIgG) were used as negative controls. **f** Meta-analysis of the averaged DOT1L ChIP-seq signal of genes across a ±10 kb genomic region flanking the TSS. **g** Normalized C/EBPβ and DOT1L ChIP-seq signal of the representative C/EBPβ-DOT1L co-targeted genes (EREG and SLC38A1). **h** Analysis of DOT1L ChIP-qPCR (green), H3K79me2/me3 ChIP-qPCR (blue), and RT-qPCR (purple) in C13* cells. Gene names in red indicate documented cisplatin-resistance genes in ovarian cancer. **i** The protein levels of C/EBPβ-DOT1L co-targeted genes in the indicated OV2008 cells were detected by western blotting. Uncropped images of blots are shown in Supplementary Figure 25. *P < 0.05; ***P < 0.001

EBPβ binding (red bars in Fig. 4h and Supplementary Figure 16a) and decreased H3K79 methylation (blue bars in Fig. 4h and Supplementary Figure 16a) of these genes in C/EBPβ-knockdown cells and the lack of change in several negative sites. When compared to mRNA levels in the control group, we found using quantitative reverse transcription polymerase chain reaction (RT-qPCR) that 25 genes were downregulated because of C/EBPβ knockdown (purple bars in Fig. 4h and Supplementary Figure 16b). Similar results were obtained in SKOV3 cells (Supplementary Figure 16c) and Caov4 cells (Supplementary Figure 16d). Taken together, our results robustly demonstrate that C/EBPβ promotes H3K79 methylation to reprogram gene expression.

**C/EBPβ modulates the function of DOT1L.** Since DOT1L is the only known H3K79 methyltransferase, and C/EBPβ is a DNA-binding protein without catalytic capability[28], we investigated whether C/EBPβ modulates the effects of DOT1L-mediated H3K79 methylation. ChIP-seq analysis showed a strong correlation between C/EBPβ- and DOT1L-targeted genes (Fig. 5a and Supplementary Data 7) and, moreover, many C/EBPβ binding sites were extremely close (<2500 bp between peak centers) to those of DOT1L (Fig. 5b). Using ChIP-qPCR analysis, we confirmed that C/EBPβ and DOT1L could bind to the same regions of genes (Fig. 5c and Supplementary Figure 17a), suggesting a protein–protein interaction between the two factors. Indeed, C/EBPβ and DOT1L were shown to directly interact with each other in co-immunoprecipitation assays (Fig. 5d) and to co-occupy the

same loci of genes in sequential and reciprocal ChIP-reChIP assays (Fig. 5e and Supplementary Figure 17b). Moreover, knocking down C/EBPβ expression resulted in a general decrease in DOT1L binding capacity in the proximal regions flanking the TSS (Fig. 5f), a finding also represented by a genome-browser view of the ChIP-seq signals (Fig. 5g) and confirmed by ChIP-qPCR analysis (green bars in Fig. 5h and Supplementary Figure 17c). Consistent with these findings, we found that DOT1L shRNA suppressed H3K79 methylation (blue bars in Fig. 5h and Supplementary Figure 17d), blocked expression of C/EBPβ-DOT1L co-targeted genes (purple bars in Fig. 5h and Supplementary Figure 17e), and abolished the effects of C/EBPβ in promoting the expression of these genes (Fig. 5i and Supplementary Figure 18). Similar results were obtained in SKOV3 cells (Supplementary Figure 19). These results demonstrate that C/EBPβ interacts with and enhances the DNA-binding activity of DOT1L to regulate H3K79 methylation.

**The effects of C/EBPβ is mediated by DOT1L.** To determine whether C/EBPβ-mediated cisplatin resistance was dependent on the effect of C/EBPβ in regulating H3K79 methylation, we inhibited DOT1L expression by transfecting DOT1L shRNAs into tumor cells or blocked DOT1L activity using the small-molecule inhibitors SGC0946 and EPZ004777[34]. We found that SGC0946 and EPZ004777 inhibited expression of DOT1L-targeted genes (MEIS1 and NANOG) in a dose-dependent manner, a finding which confirms the inhibitory effect of DOT1L activity in ovarian cancer cells (Supplementary Figure 20a). Administration of

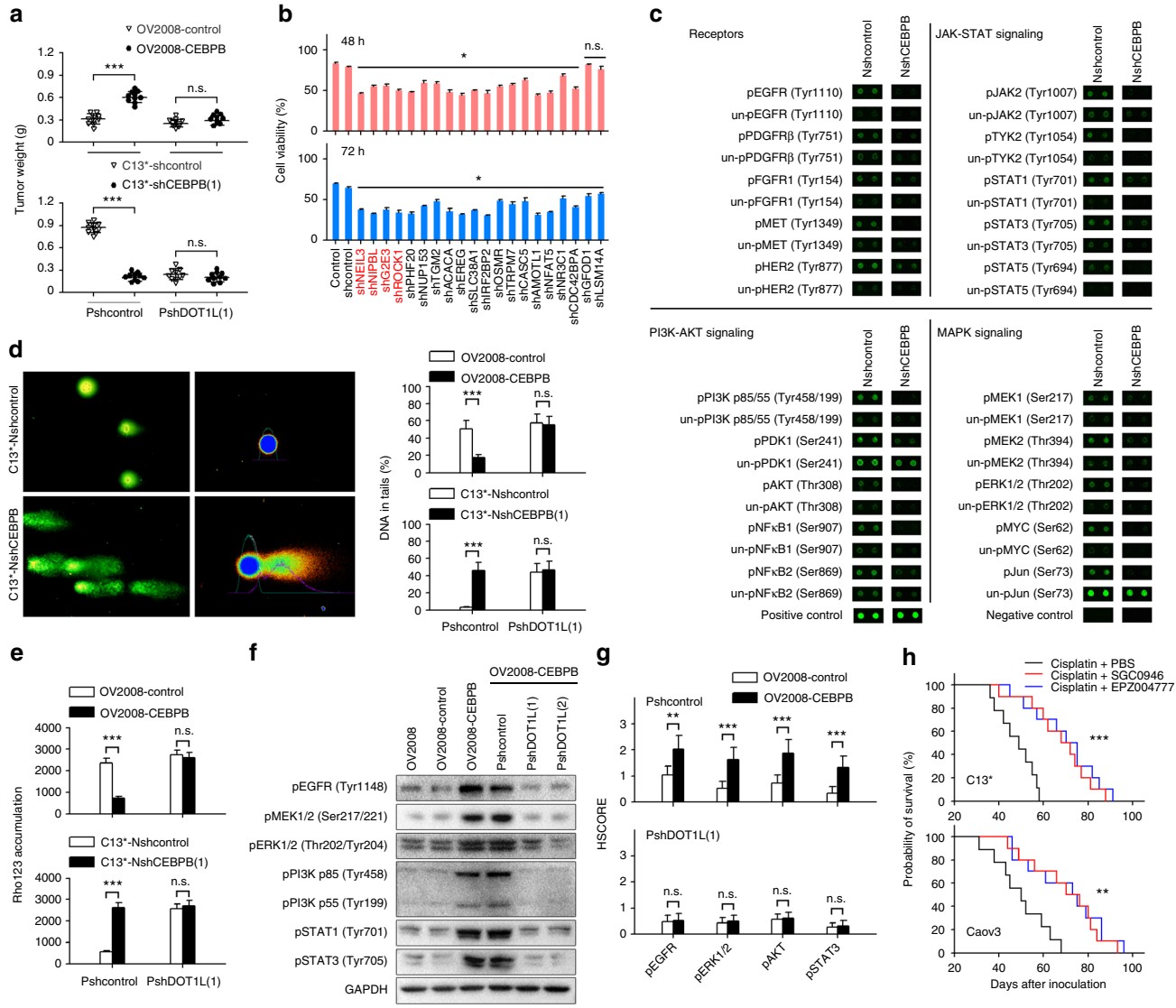

**Fig. 6** The effects of C/EBPβ on cisplatin resistance are mediated by DOT1L. **a** One week after orthotopic inoculation with the indicated cells, the mice were treated with cisplatin (5 mg/kg) intraperitoneally every 4 days; for DOT1L inhibitor treatment, 4 mg/kg SGC0946 or EPZ004777 were injected intraperitoneally every 4 days (n = 10 per group). Tumors were excised and weighed 6 weeks after tumor inoculation. **b** Assay of cisplatin-resistance genes. Genes were individually silenced by shRNA lentiviral particles in SKOV3 cells, and cell viability at 48 h and 72 h after treatment with 30 μM cisplatin was determined using CCK8 assays. **c** Phospho-antibody array. See Supplementary Figure 21 and Supplementary Data 7 for full results and additional annotations. **d** Alkaline comet assays were performed after treatment with 50 μM cisplatin for 24 h. **e** Cellular accumulation of rhodamine 123 (Rho123) was determined by flow cytometric analysis after treatment with 1 μM Rho123 for 1 h. **f** The indicated cells were treated with 50 μM cisplatin for 24 h, and protein phosphorylation was detected by western blotting. **g** IHC analysis of phosphorylated EGFR, ERK1/2, AKT, and STAT3 in xenograft tumor sections collected from mice treated with cisplatin (n = 10 per group). **h** The mice were treated as described above in **a**. The mice were maintained until death, and Kaplan–Meier survival curves of mice were plotted. Uncropped images of blots are shown in Supplementary Figure 26. *P < 0.05; **P < 0.01; ***P < 0.001. n.s. not significant

DOT1L shRNAs and inhibitors abolished the effects of C/EBPβ overexpression on promoting cisplatin resistance of OV2008 cells in vitro (Supplementary Figure 20b) and in vivo (Fig. 6a and Supplementary Figure 20c). Similarly, when DOT1L was knocked down or blocked, knockdown of C/EBPβ expression did not further reduce cisplatin resistance in C13* cells (Fig. 6a and Supplementary Figure 20).

To further clarify whether DOT1L mediates the effects of C/EBPβ on the cisplatin-resistance phenotype, we examined the downstream effects of C/EBPβ-DOT1L co-targeted genes, and found that many of these co-targeted genes were involved in cisplatin resistance. To confirm this finding in ovarian cancer, cell viability in response to cisplatin treatment was tested after genes

were individually silenced by a pool of three nonoverlapping shRNAs in SKOV3 cells. In addition to the 9 known cisplatin-resistance genes, our analysis identified 14 novel genes that contributed to cisplatin resistance in ovarian cancer cells (Fig. 6b and Supplementary Data 8). These 23 genes have the potential to promote cisplatin resistance through various pathways, falling broadly into three categories: (1) promoting drug transduction, (2) enhancing DDR, and (3) regulating signal transduction to promote cell viability (Supplementary Figure 15). Consistent with these findings, knockdown of C/EBPβ resulted in the broad suppression of JAK-STAT (Janus kinase/signal transducer and activator of transcription), PI3K-AKT(phosphatidylinositol-3-kinase/AKT), and MAPK signals (Fig. 6c, Supplementary

Figure 21, and Supplementary Data 9), and knocking down either C/EBPβ or DOT1L similarly elevated intercellular drug accumulation and enhanced cisplatin-induced DNA damage (Supplementary Figure 22). Moreover, we found that DOT1L shRNAs and inhibitors abolished the effects of C/EBPβ overexpression on the promotion of DDR (Fig. 6d and Supplementary Figure 23a), drug transduction (Fig. 6e and Supplementary Figure 23b), and survival-related signals (Fig. 6f, g and Supplementary Figure 23c). Furthermore, when DOT1L was knocked down or blocked, knockdown of C/EBPβ expression did not further reduce DDR or drug transduction in C13* cells (Fig. 6d, e and Supplementary Figure 23). Similar results were obtained in SKOV3 cells (Supplementary Figure 24). Finally, we found that combining DOT1L inhibitors with cisplatin treatment significantly extended the survival of mice inoculated with C13* or Caov3 cells compared to cisplatin alone (Fig. 6h). Taken together, these results robustly demonstrate that the promotion of cisplatin resistance by C/EBPβ is mediated by DOT1L.

## Discussion

Global changes in the epigenetic landscape are a hallmark of cancer[35]. Epigenetic alterations are nonrandom and tend to occur repeatedly at specific genomic regions during tumor progression and the acquisition of chemoresistance[36,37]. Various studies have led to the hypothesis that global epigenetic changes are controlled in a site-specific manner; however, much is still unknown about the mechanisms underlying epigenetic modulation of related groups of genes[15,36]. The results herein indicate that a set of functionally related genes involved in epigenetic reprogramming can be controlled by specific TFs. Most epigenetic enzymes including DOT1L lack the intrinsic capacity for gene-specific regulation[38]. We showed that C/EBPβ acted as a cofactor for DOT1L and colocated to target genes, thereby maintaining an open chromatin state at multiple drug-resistance genes. Therefore, the activities of TFs such as C/EBPβ mediate important mechanisms through which epigenetic enzymes modify the chromatin in a context-dependent manner. Our findings provide a rationale for developing therapeutic agents targeting C/EBPβ or the C/EBPβ-DOT1L interaction, and highlight the importance of identifying chromatin-modifying TFs in cancer.

Although genetic changes cause drug resistance, this resistance pattern was rarely observed following chemotherapy and was long thought to be permanent and unalterable[9,10]. In contrast, active mechanisms are required in the maintenance of epigenetic states, and thus the reversibility of epigenetics constitutes an attractive therapeutic target[39]. Alterations in epigenetic regulators such as enhancer of zeste 2 (EZH2, a regulator of H3K27 methylation) have a global influence on both tumorigenesis and drug sensitivity in various tumor models[40–42]. This effect likely occurs through large-scale gene modulation, in contrast to the mutation, amplification, or rearrangement of a single gene[9,36]. We discovered that H3K79 methylation plays a significant role in the progression of HG-SOC, and identified C/EBPβ as a novel regulator of histone methylation that modulates the H3K79 methylation of multiple drug-resistance genes. Inhibiting or knocking down expression of the H3K79 methyltransferase DOT1L abolished the promotional effect of C/EBPβ on the expression of drug-resistance genes and reversed cisplatin resistance in vitro and in vivo. Therefore, facilitating H3K79 methylation is an important mechanism by which C/EBPβ promotes cisplatin resistance of tumor cells.

In addition, we found that the promotional effects of cisplatin resistance by C/EBPβ were mediated by a variety of downstream genes, generally involved in three functions: drug transport (the multidrug resistance-associated proteins MRP1 and MRP3, also known as ABCC1 and ABCC3, respectively), DDR, and cell survival (Supplementary Figure 15). The generation of DNA lesions followed by activation of the DNA damage response is the major mode of action of cisplatin[43]. Of note, DNA repair signaling was the most significantly impaired pathway in C/EBPβ-knockdown cells. By promoting H3K79 methylation of targeted genes, C/EBPβ-DOT1L upregulated the expression of several key factors in homologous recombination repair, nonhomologous end-joining (e.g., NUP153, NIPBL)[44,45], base excision repair (NEIL3)[46], and protection against DNA damage-induced apoptosis (PHF20 and TGM2)[47,48]. Correspondingly, targeting C/EBPβ-DOT1L by shRNA or small-molecule inhibitors significantly enhanced cisplatin-induced DNA damage and cell apoptosis. BRCA1 is a key component of the homologous-recombination double-strand DNA repair pathway. Decreased BRCA1 expression is associated with tumorigenesis of ovarian cancer, but is also associated with platinum sensitivity and better prognosis[49,50]. Although C/EBPβ did not regulate BRCA1 expression, knocking down C/EBPβ led to decreased expression of many genes, such as BRCA2, ATM, and EGFR, in the BRCA1-related DNA damage response pathway. IPA also predicted suppression of BRCA1-related DDR in C/EBPβ knockdown cells. Our data indicate that C/EBPβ expression may play an important role in BRCA1/2-related DNA damage responses. Further investigation of the relationship between C/EBPβ and other important factors (e.g., BRCA1/2, CCNE1) in HG-SOC may provide important insights into this disease.

We also discovered that C/EBPβ-DOT1L had a broad effect on survival signals. In the face of cisplatin treatment, targeting C/EBPβ-DOT1L resulted in reduced phosphorylation levels of ERK1/2, PI3K-AKT, and JAK-STATs. Cisplatin-resistance genes, including RICTOR (AKT-mTOR), SLC38A1 (AKT), OSMR (STAT3), LDLR (ERK), and HIF1A (a multidrug resistant gene coordinated with EGFR signals), may participate in C/EBPβ-induced cisplatin resistance by regulating an individual pathway. Moreover, we identified several C/EBPβ downstream genes (AMOTL1, CASC5, GFOD1, LSM14A, NFAT5, NR3C1, and TRPM7) that, to our knowledge, are new cisplatin-resistance genes, whose functions remain to be elucidated. In the current study, we illustrated the relevance of downstream genes in C/EBPβ-mediated signaling; however, considering the large number of downstream genes and inherent network complexity, more research is needed to understand the precise roles of these and possibly other genes involved in C/EBPβ-induced cisplatin resistance.

C/EBPβ plays important roles in the regulation of genes to control differentiation, inflammation, metabolism, cell survival, and oncogene-induced senescence[51–53]. Our results demonstrate that C/EBPβ reprograms H3K79 methylation to enhance chemoresistance of ovarian cancer. However, the mechanisms by which C/EBPβ is upregulated in ovarian cancer are still unknown. The transcriptional activity of C/EBPβ is held in an intrinsically repressed state by several regulatory regions. Phosphorylation or deletion of these inhibitory domains leads to increased transcriptional activity of C/EBPβ[30]. The C/EBPβ promoter contains binding sites for several TFs, including C/EBPβ itself, STAT3, specificity protein 1 (Sp1), members of the CREB/ATF family, EGR2, Fra-2, SREBP1c, Myb, and RARA[51]. Genomic alterations may also affect C/EBPβ expression; gain of the chromosomal region containing CEBPB is associated with lobular carcinoma in situ of the breast[30,54]. Moreover, there is increasing evidence that C/EBPβ is primarily regulated via post-transcriptional mechanisms, and that mRNA levels are not necessarily regulated[30]. The results herein suggest that changes to C/EBPβ after platinum exposure may be primarily because of post-transcriptional mechanisms, while the transcriptional regulation of C/EBPβ was involved in the tumorigenesis of HG-SOC.

Further research is warranted to understand the mechanisms for C/EBPβ upregulation in both tumorigenesis and after chemotherapy.

However, this study had several limitations. In particular, the provenance of many 'high-grade serous' ovarian cancer cell lines has recently been questioned[55,56]. SKOV3, an originally putative serous cell line, harbors mutations in TP53, PIK3CA, ARID1A, and gene amplification in ERBB2 and CCNE1 loci, which are often identified in high-grade 'ovarian' endometrioid tumor or in the 'uterine' serous tumor. There is a likelihood that SKOV3 initially derived from either one of these histological types rather than from the 'ovarian' serous type[57–59]. The cisplatin-sensitive and -resistant cell lines pair OV2008 and C13* were originally established from a patient with serous cystoadenocarcinoma of the ovary, but have not yet been classified using high-throughput technology similar to that used by Domcke et al.[55] Nevertheless, OV2008 and C13* were good model for investigating chemoresistance in the present study, because they represented the shift in C/EBPβ expression upon cisplatin treatment. In addition, there was a technical limitation associated with ChIP-Seq-related experiments[60]. ChIP-Seq was not performed in triplicate and ChIP binding events were not validated by another antibody specific to C/EBPβ or DOT1L.

In summary, we demonstrated that specific TFs were responsible for regulating chromatin modifications, and that an aberrant epigenetic program involving histone methylation drove cisplatin resistance. By directly interacting with DOT1L, C/EBPβ-mediated reprogramming of gene expression triggered a broad signal network that synergized to promote cisplatin resistance. These results propose a new path against cancer epigenetics in which identifying and targeting the key regulators of epigenetics such as C/EBPβ may provide more precise therapeutic options in ovarian cancer.

## Methods

**Cells**. OV2008 and C13* cell lines were gifts from Professor Benjamin K. Tsang of the Ottawa Health Research Institute, Ottawa, Canada[61]. The cisplatin-sensitive ovarian cancer cell line OV2008 was originally established from a patient with serous cystoadenocarcinoma of the ovary, and the cisplatin-resistant C13* cells were generated from OV2008 cells by monthly in vitro selection with cisplatin[62,63]. The cells were maintained in RPMI-1640 (cat. no. 31800-089; Gibco/Invitrogen, Carlsbad, CA, USA) supplemented with 2 mM L-glutamine and 10% fetal bovine serum (FBS; cat. no. 10438-026; Gibco/Invitrogen). SKOV3, ES-2, Caov3, Caov4, and OV-90 cells were purchased from the American Type Culture Collection (Rockville, MD, USA) in December 2013 and cultured according to the manufacturer's guidelines. All of the cell lines were routinely checked for mycoplasma contamination (Mycoalert Mycoplasma Detection Kit, Lonza) and were authenticated by their source organizations prior to purchase. All cells used for the experiments were passaged less than 20 times.

**Cell transfection**. Tumor cells were transfected with CMV-luciferase-IRES-RFP lentiviral particles (GeneChem, Shanghai, China) and isolated by fluorescence-assisted cell sorting (FACS) for live-cell imaging. For C/EBPβ overexpression, cells were transfected with sv40-Neomycin-CMV-CEBPB lentiviral particles, constructed by GeneChem Co., Ltd. ShRNA lentiviral particles (GeneChem) were used to knock down the expression of C/EBPβ and DOT1L in tumor cells. The sequences targeted by shRNAs were as follows: NshCEBPB(1), 5′-TGCCTTTAAATCCATGGAA-3′; NshCEBPB(2), 5′-ACTTCCTCTCC-GACCTCTT-3′; PshDOT1L(1), 5′-GAGTGTTATATTTGTGAAT-3′; PshDOT1L (2), 5′-CACCTCTGAACTTCAGAAT-3′. 'N' and 'P', the first letter in the designations for the shRNAs, indicate whether the vector harbors a neomycin- or puromycin-resistance gene, respectively. Nshcontrol and Pshcontrol, which did not target any known gene, were used as controls. After selection with G418 and/or puromycin, cells with stable transfection of shRNA were used for subsequent experiments.

**Reagents and antibodies**. SGC0946 and EPZ004777 were purchased from Selleck Chemicals (TX, USA). Antibodies used in ChIP were as follows: anti-H3K4me3 (ab8580; Abcam, CA, USA), anti-H3K9me3 (ab8898; Abcam), anti-H3K27me3 (07-449; Millipore, MA, USA), anti-H3K36me3 (ab9050; Abcam), anti-H3K79me2/me3 (ab2621; Abcam), anti-H4K20me3 (ab9053; Abcam), anti-C/EBPβ (ab18336; Abcam), and anti-DOT1L (ab72454; Abcam). Primary antibodies used in western blot and IHC were as follows: anti-C/EBPβ (SAB4500112; Sigma, St. Louis, MO, USA), anti-DOT1L (ab72454; Abcam), anti-RICTOR (GTX104617; GeneTex, USA), anti-NEIL3 (11621-1-AP; Proteintech, Wuhan, China), anti-NIPBL (18792-1-AP; Proteintech), anti-PHF20 (ab157192; Abcam), anti-SLC38A1 (12039-1-AP; Proteintech), anti-AMOTL1 (ab171976; Abcam), anti-pEGFR (Tyr1148; cat. no. #4404; Cell Signaling Technology, Danvers, MA, USA), anti-MEK1/2 (Ser217/221; cat. no. #9154; Cell Signaling Technology), anti-pERK1/2 (Thr202/Tyr204; cat. no. #4370; Cell Signaling Technology), anti-pPI3K p85 (Tyr458)/p55 (Tyr199; cat. no. #4228; Cell Signaling Technology), anti-pAKT (Thr308; cat. no. #13038; Cell Signaling Technology), anti-pSTAT1 (Tyr701; cat. no. #7649; Cell Signaling Technology), anti-pSTAT3 (Tyr705; cat. no. #9145; Cell Signaling Technology), and anti-GAPDH (10494-1-AP; Proteintech).

**Magnetic separation and preparation of pooled samples**. Because tissues are composed of a mixture of different cells with distinct epigenetic backgrounds, we purified the cells from freshly collected samples using beads coated with epithelial cell target antibodies. Briefly, tissues were minced into small fragments and digested with collagenase I (1 mg/mL in RPMI-1640 supplemented with 10% FBS) at 37 °C for 40 min. The single-cell suspension was then incubated with 100 μL of EpCAM MicroBeads (cat. no. 130-061-101; Miltenyi Biotec, Auburn, CA) per 5 × 10^7 cells for 30 min at 4–8 °C. Epithelial cells that bound to the beads were separated by a magnet, and sorting was repeated.

The samples using in high-throughput sequencing study (Fig. 1) included 20 HG-SOC samples (Supplementary Table 1) and the fimbriae of fallopian tubes from 20 patients with benign diseases, including 10 patients with hysteromyoma and 10 patients with adenomyosis, underwent hysterectomy with bilateral salpingectomy. There was no statistical difference in age between the two groups ($P = 0.161$, analysis of variance (ANOVA)). The hematoxylin and eosin staining of fallopian tubes were reviewed by two pathologists to confirm no significant pathologic changes. To overcome the heterogeneity and relatively large variability between patients, we used pools of samples for the HG-SOC and normal control groups[64,65]. Pooled results were then confirmed in individual patient analysis. Briefly, samples were individually sorted by magnetic beads as described above. Then, the purified samples were individually subjected to RNA extraction and some ChIP steps (from formaldehyde crosslinking to ultrasonic chromatin fragmentation) as described above. After determining the RNA/DNA concentrations using a Nanodrop (Thermo), equal amounts of RNA were pooled to a total of 2 μg per group and then used for library construction for RNA-seq; equal DNA amounts of fragmented chromatins were pooled to a total of 200 μg per group and then subjected to the remaining steps in the ChIP experiment (from preclearing with protein A/G beads to ChIP-DNA purification). The remaining RNA and fragmented chromatin were also individually stored at −140 °C for individual patient analysis. For data verification in individual patients, we randomly selected 50 differentially expressed genes and another 50 genes with different histone methylation states (HG-SOC vs. normal control groups). Importantly, 46/50 (92%) and 43/50 (86%) genes were verified by RT-qPCR and ChIP-qPCR, respectively, demonstrating that the results from pooled samples had acceptable reliability.

**ChIP and ChIP-reChIP**. ChIP was performed with an EZ ChIP Chromatin Immunoprecipitation Kit (cat. no. 17–371; Millipore) according to the manufacturer's protocol. Briefly, 2 × 10^7 crosslinked cells were lysed in ChIP lysis buffer (1% sodium dodecyl sulfate (SDS), 10 mM ethylenediaminetetraacetic acid (EDTA), and 50 mM Tris, pH 8.1) containing protease inhibitor cocktail (cat. no. 04 693 132 001; Roche). The chromatin was fragmented to 200–400 bp using a sonicator (Sonics, USA). Equal amounts of DNA were diluted and precleared with protein A/G beads for 1 h at 4 °C. After repeating DNA quantification, 1% of the sample (as input) was saved at 4 °C, and the remaining sample was incubated with 5 μg antibody and protein A/G beads overnight at 4 °C. After washing, the beads were resuspended in elution buffer (1% SDS, 100 mM NaHCO₃) and subjected to RNase A and proteinase K digestion. Crosslinking was then reversed at 65 °C for 8–10 h. DNA was recycled with a DNA purification kit (Qiagen, Valencia, CA, USA). The purified ChIP-DNAs were analyzed by qPCR on a CFX96 Touch Real-Time PCR Detection system (Bio-Rad Laboratories, Inc., CA, USA) using iQ SYBR Green Supermix (Bio-Rad Laboratories, Inc.) and reported as the percentage of input. The primer sequences using in ChIP-PCR are shown in the Supplementary Data 10. In ChIP-reChIP, immunocomplexes from the first ChIP were eluted with 10 mM dithiothreitol in Tris-EDTA buffer at 37 °C. The eluates were then diluted with 10 volumes of ChIP lysis buffer and used for the second round of ChIP.

**ChIP-seq**. The purified ChIP-DNAs were subjected to library construction for ChIP-seq. The library construction and sequencing procedures were performed by Generay Biotechnology (Shanghai) Co., Ltd, following the manufacturer's instructions (Illumina, San Diego, CA, USA). Briefly, ChIP-DNA quality was examined by Qubit (Invitrogen) using a Quant-iT PicoGreen double-stranded DNA Assay Kit (Life Technologies), and input DNA was examined by gel electrophoresis. DNAs were prepared for end repair and 'A' tailing, adaptor ligation, and library amplification using a TruSeq chip DNA LT Sample Prep Kit (Illumina). The ligation products were purified and accurately size selected (200–400 bp) by

agarose gel electrophoresis. This size-selection step was repeated after PCR amplification with DNA primers (Illumina). Next, 50-bp single-end sequencing was conducted using the Illumina Hiseq 2000 platform for tumor samples and the Illumina Hiseq 2500 platform for cell line samples.

**Bioinformatics analysis of ChIP-seq data.** Quality control of ChIP-seq data was performed using FastQC, and then data were mapped to human genome hg19 using Bowtie2. The identification of ChIP-seq peaks (bound regions) was performed using a custom approach (HOMER) (http://homer.ucsd.edu/homer/). All parameters were applied at the default setting. The $P$value cutoff for the peak detection was $10^{-4}$. The input group was used as a control. The results were visualized with IGV software. The peak information was annotated with PeakAnalyzer. For comparisons of ChIP-Seq tag densities between different sequencing libraries, all ChIP-Seq profiles were normalized to $10^7$ total tag numbers. Overlapped peaks were modeled using the least squares method, and the $P$ value of peak-difference analysis was then calculated using a Bayesian model. De novo motif analysis was performed using the HOMER software package; peak sequences were compared with 50,000 randomly selected genomic fragments of the same size, and all the parameters were applied at the default setting.

**RT-qPCR and RNA-seq.** Total RNA was routinely extracted using TRIzol reagent (Invitrogen; Thermo Fisher Scientific, Inc.), and RNA quality was examined by Nanodrop (Thermo, Waltham, MA, USA) analysis and gel electrophoresis. The relative quantity of mRNA was determined by RT-qPCR using a CFX96 Touch Real-Time PCR Detection system (Bio-Rad Laboratories, Inc.) with iQ SYBR Green Supermix (Bio-Rad Laboratories, Inc.). The expression levels of genes were quantified using the comparative $C_T$ method. The expression level of each mRNA was normalized to the level of *GAPDH* mRNA and expressed as the fold difference relative to the control. The primer sequences used in RT-qPCR are shown in the Supplementary Data 11. RNA library construction and sequencing were performed by Generay Biotechnology (Shanghai) Co., Ltd, following the manufacturer's instructions (Illumina). Briefly, ribosomal RNA was removed from total RNA using TruSeq Stranded Total RNA with Ribo-Zero for Humans (Illumina). First-strand complementary DNA (cDNA) was then synthesized using random hexamer-primed Superscript II Reverse Transcriptase (Invitrogen), followed by second-strand cDNA synthesis using RNase H and DNA polymerase and ligation of sequencing adapters using a TruSeq RNA LT Sample Prep Kit v2 (Illumina). Then, 50-bp single-end sequencing was conducted using an Illumina Hiseq 2000 platform for tumor samples or an Illumina Hiseq 2500 platform for cell line samples.

**Bioinformatics analysis of RNA-seq data.** Sequence data quality check was performed using FastQC. The RNA-Seq data were mapped to the hg19 reference genome by TopHat for Illumina using default options. Assembly of transcripts and estimation of their abundance (fragments per kilobase of exon per million fragments mapped (FPKM)) were carried out using Cufflinks software. Differential gene expression analyses between groups were performed using HTSeq software for tumor samples[66] and DESeq2 software for cell line samples[67]. Heatmap.2 in the 'gplots' package of the R program was used for the construction of heat maps. Genes that were up- or downregulated in both C13* and SKOV3 cells with a $P$ value of less than 0.05 were analyzed using IPA software (Qiagen, Redwood City, CA, USA; http://www.ingenuity.com) in order to assign the genes to different functional networks. Fisher's exact test was utilized to calculate $P$ values with IPA. IPA generated a $z$-score for each predefined canonical pathway, where a $z$-score of at least 2 was associated with a confidence level of at least 99% that results were not chance. Positive and negative $z$-scores represented the activated and suppressed states, respectively.

**Whole-exon sequencing.** Genomic DNA was extracted using the Tissue DNA Kit (Omega Bio-Tec, USA). The qualified genomic DNA was randomly fragmented by Covaris technology and the size of the library fragments was mainly distributed between 150 bp and 250 bp. DNA fragments were end-repaired, ligated with adapters, and amplified. Each resulting qualified captured library with the SureSelect Human All Exon kit (Aglient) was then loaded on BGISEQ-5000 sequencing platforms, and we performed high-throughput sequencing for each captured library. High-quality reads were aligned to the human reference genome (GRCh37) using the Burrows–Wheeler Aligner (BWA v0.7.15) software. All genomic variations, including single-nucleotide polymorphisms and InDels were detected by HaplotypeCaller of GATK (v3.0.0).

**Clinical samples.** Tumor samples from a total of 277 patients with epithelial ovarian cancer (including 245 serous, 18 mucinous, 8 clear-cell, and 6 endometrioid ovarian cancer) were obtained from Clinical Database and Biobank of Patients with Gynecologic Neoplasms, under ClinicalTrials.gov Identifier NCT01267851 (ethical approval is available at https://clinicaltrials.gov/ct2/show/study/NCT01267851). Pathology review was performed by two pathologists. Informed consent was obtained from all patients. The sample sizes were chosen based on the results of a power analysis. All patients underwent operation and 6–8 courses of postoperative platinum-based chemotherapy. Patient characteristics are summarized in Supplementary Table 2. OS was calculated from the date of initial

surgery to the date of last known contact or death. PFS was calculated from the date of initial surgery to the date of progression/recurrence or last known contact if the patient was alive and had not experienced recurrence. Platinum resistance and platinum sensitivity were defined as progression/relapse within 6 months and after 6 months from the last platinum-based chemotherapy, respectively[25,68]. At the time of analysis, for SOC, 84 (34.3%) of the 245 SOC patients had died, and 140 (68.3%) of the 205 evaluable patients had experienced disease progression, resulting in a median OS of 50.47 months (95% confidence interval (CI), 29.7–71.3 months) and median PFS of 27.1 months (95% CI, 22.4–31.9 months). The average length of follow-up for the 161 patients still alive was 33.3 months (range, 6.8–108 months).

**Analysis of prognosis using TCGA dataset.** The gene expression data (Affymetrix U133A platform) for 489 cases of HG-SOC were downloaded from TCGA data portal for analysis of prognosis (http://cancergenome.nih.gov/); the corresponding clinical information is publicly available[25]. Normalization algorithms were used to transform sample signals to minimize the effects of variations arising from nonbiological factors. To improve comparability of the results, samples were divided into low- and high-expression groups based on the median of each gene.

**Gene expression meta-analysis.** The gene expression meta-analysis in patients with ovarian cancer was performed using the 'curatedOvarianData_1.3.5' Bioconductor package, which contains uniformly prepared microarray data for 4411 patients from 30 studies with curated and documented clinical metadata[29]. The follow-up threshold was limited to 7 years, and all other parameters were defaults. Hazard ratios indicated the factor by which overall risk of death increased with a one standard deviation increase in gene expression.

**Primary culture.** Primary derivative cultures of ovarian cancer cells were performed as described previously, using freshly isolated ascites fluids from patients with SOC[69]. Briefly, freshly isolated ascites fluids from SOC patients were transferred to tissue culture flasks in sterile condition, adding equal volume of complete MCDB/M199 medium supplemented with 100 U/mL penicillin and 100 mg/mL streptomycin. Then, the isolated ascites fluids were placed in an incubator undisturbed for 3–4 days prior to the first change of complete medium, and media were changed every 2–3 days until the flasks were confluent. After 1–2 weeks of continuous culture, contamination with unwanted cell types, such as fibroblasts and hematopoietic cells, was rarely observed. The cells were then used in following experiments if the cell purity exceeded 95%, as verified by flow cytometry using the epithelial marker EpCAM. Cells were used at passages 2–3.

**Animal studies.** Female NOD-SCID mice (4 weeks old) were purchased from Beijing HFK Bio-Technology Co. Ltd (Beijing, China). All animal experiments were approved by the Committee on Ethics of Animal Experiments of Tongji Medical College. The mice were maintained in an accredited animal facility at Tongji Medical College. Animal numbers were determined based upon the results of a power analysis in combination with previous experience to provide 80% power for a test at a significance level of 0.05. The mice were assigned randomly to each group. An orthotopic model of ovarian cancer was established as we described previously[46]. Briefly, $1.0 \times 10^6$ cells in 10 μL serum-free growth medium were injected under the ovarian bursal membrane. Tumor growth was dynamically monitored in living mice by optical imaging of luciferase activity using the IVIS SPECTRUM system (Caliper, Xenogen USA). Tumor volume ($mm^3$) was measured by three-dimensional image reconstruction using Living Image software version 4.3.1. To exclude the change in proliferation rate upon modulation of C/EBPβ protein levels, cisplatin-induced tumor reduction rate was calculated based on the tumor volume in PBS-treated group at each time point, using the following formula: reduction rate (%) = $(1 - V_{cisplatin}/V_{PBS}) \times 100$, where $V_{cisplatin}$ is the tumor volume in cisplatin-treated group, and $V_{PBS}$ is the average tumor volume in PBS-treated group at the same time point. To measure tumor weight, the mice were killed at 6 weeks after tumor cell inoculation, and their tumors were excised. Investigators were blinded to the treatment groups.

**IHC.** Formalin-fixed, paraffin-embedded tissue sections were subjected to immunohistochemical analysis using an Avidin-Biotin Complex (ABC) Vectastain Kit (Zsgb-Bio, Beijing, China) according to the manufacturer's protocols. Fixed positive and negative controls were evaluated in each experiment to control for staining variability among batches of experiments. For semiquantitative evaluation of protein levels in tissues, an immunoreactivity-scoring system (HSCORE, range from 0 to 3) was used[70]. Briefly, the staining intensity was graded (0, absence; 1, weak; 2, moderate; 3, strong). The HSCORE was calculated using the following formula: HSCORE = $\sum Pi \times i$, where $i$ is the staining intensity of tumor cells, and $Pi$ is the percentage of corresponding cells at each level of intensity. HSCORE <1.5 was classified as a low protein level, and HSCORE ≥1.5 was classified as a high protein level. Each data point represents the mean score of two pathologists, who were blinded to all clinicopathologic variables.

**Co-immunoprecipitation.** Co-immunoprecipitation (co-IP) was performed using the Pierce Crosslink Immunoprecipitation Kit (Thermo Scientific) as described

previously[70]. Mouse anti-human C/EBPβ (cat. no. ab18336; Abcam) and rabbit anti-human DOT1L (cat. no. ab72454; Abcam) antibodies were used for IP. Normal mouse IgG and rabbit IgG were used as controls.

**Western blotting**. Western blotting was carried out as described previously[71]. The signals were detected using a SuperSignal West Pico Chemiluminescent Substrate kit (Pierce Biotechnology, Inc., IL, USA) on a ChemiDoc XRS+ machine (Bio-Rad Laboratories). Glyceraldehyde 3-phosphate dehydrogenase (GAPDH) was employed as a loading control. The relative expression level of proteins was analyzed using Image Lab Software (Bio-Rad Laboratories).

**Phospho-antibody array**. C13*-shcontrol and C13*-shCEBPB cells were treated with 50 μM cisplatin for 24 h. The cell lysates were then obtained and applied to a Cancer Signaling Phospho-Antibody Array (PCS248; Full Moon Biosystems, CA, USA). The array experiment was performed by Wayen Biotechnologies (Shanghai) Inc. according to the manufacturer's protocol. The array contained 269 site-specific and phospho-specific antibodies relative to 97 proteins, each of which had 6 replicates. The slide was scanned on a GenePix 4000B scanner (Axon Instruments, USA), and the images were analyzed with GenePix Pro 6.0. The fluorescence intensity of each array spot was quantified, and the mean value was calculated. The following equation was used to calculate the phosphorylation signal ratio: phospho ratio = phosphorylated A/unphosphorylated A, where A represents the target protein. Then, the ratio of the phospho ratio between groups was calculated using the following formula: ratio of phospho ratio = phospho ratio (C13*-shCEBPB)/ phospho ratio (C13*-shcontrol). The 95% CIs were used to quantify the precision of the phosphorylation ratio based on analysis of the replicates.

**Alkaline comet assays**. Cisplatin-induced DNA damage was evaluated by alkaline comet assays using a CometAssay Kit (cat. no. 4250-050-K; Trevigen), as described previously[49]. DNA strand breakage was expressed as 'DNA in tails (%)'. The average percentage of the DNA in tails was measured for 100 cells (at least) per sample using CometScore1.5 software (Tritek). Results were obtained from five independent experiments.

**Cell viability assays**. Tumor cells were seeded at $8 \times 10^3$ cells per well in 96-well plates. Cells were then treated for the indicated times with cisplatin. Using a cell counting kit-8 (CCK8; Boster, China), the relative quantity of the cells at each time point was measured with a Multiskan Spectrum microplate reader (μQuant Bio-Tek Instruments, USA) at a wavelength of 450 nm. The assay was performed using five replicates. Cell viability was expressed as a percentage relative to the value at day 0.

**Colony formation assays**. Cells were seeded in 6-well plates and treated with 50 μM cisplatin for 12 h. After incubating at 37 °C for 2 weeks, the cells were fixed with 4% paraformaldehyde and stained with 0.1% (w/v) crystal violet. Colonies of fifty cells or more were counted manually, and colony formation rates were expressed as the percentage of colonies in cisplatin-treated cultures compared with that in control cultures. Each assay was performed in triplicate.

**Flow cytometric analysis**. For analysis of the EpCAM-positive ratio, we used fluorescein isothiocyanate (FITC)-conjugated anti-human EpCAM antibody (cat. no. 130-080-301; Miltenyi Biotec). For apoptosis assays, cells were harvested by trypsinization and stained with annexin V-FITC and propidium iodide from an Apoptosis Detection kit (cat. no. AP101-100; MultiSciences Biotech) according to the manufacturer's instructions. Data were acquired on a FACSCalibur flow cytometer (BD Biosciences) and analyzed with CellQuest software. For analysis of the cellular accumulation of rhodamine 123 (Rho123), the percent staining was defined as the percentage of cells in the gate, which was set to exclude ~99% of isotype control cells. The fluorescence index was calculated as the mean fluorescence multiplied by the percentage of positively stained cells[72]. The experiments were repeated three times.

**Statistical analysis**. Bioinformatics analysis of ChIP-seq and RNA-seq data are described above. Results were interpreted by one-way ANOVA, unless otherwise indicated. Correlation analyses were performed using Pearson's correlation tests. Differences in patient survival were examined with Kaplan–Meier curves using the log-rank test. SPSS (version 13.0) software was used for statistical analysis. Differences with two-sided P values of less than 0.05 were considered statistically significant.

**Data availability**. The data that support the findings of this study are available from the corresponding author upon request. The ChIP-seq and RNA-seq data are available under accession number SRP076250 in the NCBI SRA database.

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

## Acknowledgements

This work was supported by the National Science Foundation of China (grant nos. 81502250, 81600448, 81472783, 81230038, 81372801, 81772787, 81572570, and 81272426), the '973' Program of China (grant no. 2015CB553903), and National Science-technology Support Projects (grant no. 2015BAI13B05). We thank Dr. Guan Wang and Huo-Jun He from Generay Biotechnology (Shanghai) Co., Ltd for assistance with bioinformatic analysis.

## Author contributions

D.L. and X.-X.Z. performed project conception, design, experimental work, data interpretation, and preparation of the manuscript. M.-C.L., C.-H.C., D.-Y.W., B.-X.X., J.-H.T. and J.W. performed experimental work. Z.-Y.Y., X.-X.F., F.Y., G.C., P.W., L.X., H.W., and J.-F.Z. participated in data interpretation. Z.-H.F. reviewed the manuscript and participated in project conception, design, and data interpretation. D.M. and Q.-L.G. supervised the study and participated in project conception, design, data interpretation, and revision of the manuscript.

## Additional information

**Competing interests:** The authors declare no competing interests.

