## [Peer Review File(PDF 304 kb) · Nature Communications]

Reviewers' comments:

Reviewer #1 (Remarks to the Author):

The authors examined the role of C/EBP β in chemoresistance in ovarian cancer. Please consider the following:

1. The Introduction is very difficult to read and should be rewritten to carefully pose the question that will be addressed in the manuscript.
2. Page 5. While a fraction of high-grade serous are now thought to arise from the fallopian tubes, there are other sites that likely serve as the site of origin. The language at the beginning of this page around this topic should be carefully checked.
3. Page 5. For epithelial cell isolations from the fallopian tubes, what kind of "benign diseases" did they have? What portion of the fallopian tube was used? Were the two groups age matched? This is important since there could be age-specific differences in methylation profiles.
4. Figure 1b and page 5. It is unclear why borderline ovarian tumors were used. Borderline tumors are thought to be more of a precursor for low-grade serous cancers, but not for high-grade serous tumors. Inclusion of borderline tumors would seem irrelevant for the present study.
5. Supplementary table 2. Contemporary cut-off for residual disease would be R0 (i.e., no gross residual) or R1 (any residual). If such data are available, then authors may want to consider presenting that.
6. Page 5 and figure 1d. The data are difficult to follow. What was the apparent strength of correlation? Such correlation values should be presented. It is important to understand such data carefully since some correlation may be statistically significant but functionally irrelevant. Also, on the figure, some of the p-values are cut off from the figure.
7. Figure 2a and page 6. What was the level of stringency for the motif analysis?
8. Figure 2. Again, it is unclear why borderline tumors are included in this analysis.
9. Figure 2b. How were the cut-off levels selected for "high" versus "low"?
10. Figure 2f. As above, what were "high" versus "low" levels?
11. Page 7. Would suggest avoiding vague terms such as "dramatic" with regard to patient outcomes. This is simply a correlational study and language suggesting causality should be avoided.
12. Page 9. Fairly limited analyses are presented for other chemotherapy drugs. It would be important to understand whether the potential effects are indeed restricted to platinum or if other commonly used chemotherapy drugs (e.g., doxorubicin, topotecan) are also affected.
13. Figure 3h. While some changes in CEBPB may be seen in patient samples after exposure to platinum, what would be important to understand is whether there were differences in PFS based on CEBPB level changes.
14. Figure 4d. Most of the gene changes appear to be relatively modest. Were there any (if so, how many) that exceeded the typical 2-fold cut-off used? What was the extent of knock-down seen for CEBPB with the siRNA?
15. Page 13. While the authors acknowledge various mechanistic possibilities, it would be more convincing if definitive mechanistic insights were gained/provided.
16. There are multiple grammatical problems that should be carefully fixed.

Reviewer #2 (Remarks to the Author):

In this manuscript, the authors present data to support a role for C/EBP β as a driver of platinum resistance and poor outcome in high grade serous ovarian cancer.

I found the manuscript dense and rather difficult to follow. However, the hypothesis is that expression of C/EBP β (via transcriptional upregulation of CEBPB) in HGSOC induces altered gene expression via an interaction with the methyltransferase DOT1L and subsequent H3K79 methylation. The altered gene expression results in more rapid cell growth and resistance to

platinum – multiple potential genes are involved in the platinum resistance. In patient samples and cohorts, increased transcription of CEBPB and expression of C/EBP β are both associated with poorer outcome and increased likelihood of relapsing in a platinum-resistant timeframe.

The manuscript presents a very amount of information. A large number of techniques have been employed across many samples, and overall this is technically impressive work. It should be noted that the first description of increased C/EBP β expression in ovarian cancer came in 1999 (Sundfelt et al Br. J. Cancer 79:1240), so this is not entirely novel. However, the data suggesting increased expression and altered histone methylation are largely convincing.

Major points

The main issue is the big picture. Currently, this manuscript is a very detailed description of biology of C/EBP β . However, the manuscript fails to address the overall landscape of ovarian/fallopian high grade serous carcinoma. This is a disease of universal TP53 mutation and extreme copy number alteration. The large consortia, in particular TCGA and ICGC, have revealed a great deal about the genomic drivers, gene expression classifiers and copy number states of these tumours. There are, in addition, many known important biological factors in this disease (e.g. germline BRCA1/2 mutation, BRCA1 promoter methylation, CCNE1 amplification, PTEN loss etc etc). Having read this manuscript several times, I still do not understand two vital pieces of information: What is the relationship of C/EBP β expression to the myriad of other important factors in HGSC, and what drives CEBPB upregulation?

Minor points

There are a series of minor technical issues that need to be addressed:

1. Identification of primary cells. The first figure in the manuscript relates to differential methylation in primary HGSOC cells vs normal "oviduct" (the term 'fallopian tube' is more widely used). However, it is difficult from the supplementary methods to understand how the fallopian tube cells were isolated and purified. Moreover, it states that primary ascites culture in MCDB/M199 medium was used to purify tumour cells – the only method for identification of these as tumour cells was EpCAM staining. However, EpCAM staining alone is insufficient. There needs to be confirmation of lineage (e.g. PAX8) and confirmation of mutant TP53 in the cultured tumour cells.
2. Clinical data. In 2005, FIGO changed the grading of serous carcinomas into a binary grading – low grade serous carcinoma and high grade serous carcinoma. The clinical data in supplementary table 1 state that some cases were grade 2. These tumours were thus either collected prior to 2005 or graded according to the old system. Pathology review should be performed.
3. Similarly, the univariate and Cox proportional hazards model uses grade as one of the analyses. However, all high grade serous carcinomas are, by definition, high grade, and so it is not possible to undertake an analysis by grade.
4. Furthermore, there are two other features of concern in these univariate/multivariate analyses. The first is age. Increased age is widely accepted to be a poor prognostic factor (as revealed in multiple analyses), but was not significant here. This may relate to the age cut-off of 50 years. Why was 50 used as the cut-off? The median age of diagnosis of ovarian cancer is over 60 (in this reviewer's centre, it is 65.2 years) and HGSOC cases aged <50 are rare (in this reviewer's centre, only 6.2% HGSOC patients were aged <50 at time of diagnosis). The second is stage. Stage is a critical prognostic factor but was again negative here. This may relate to the classifiers used (stage II and III vs IV). More appropriate is stage II vs stages III and IV, given the clinical and prognostic significance of dissemination beyond the pelvis in stage III. Overall, these analyses need to be re-performed.
5. Justification is required for the method to delineate tumours as 'CEBPB high' vs 'CEBPB low', as well as 'C/EBP β high' vs 'C/EBP β low' in Figure 2. Why were the stated cut-off values used?
6. Cell lines. The authors will be aware that the provenance of many 'high grade serous' ovarian cancer cell lines has recently been questioned (Domcke et al Nature Commun. 2013). Of the cell lines used by the authors, only CAOV3 is thought to be 'likely high grade serous' by Domcke et al. ES-2 and OV90 are 'possibly high grade serous' and SKOV3 is listed as 'unlikely'. I cannot find

details of OV2008 on either the COSMIC or CCLE databases. Reference 44 is cited to describe both OV2008 and C13*, but there is no description of the generation or characterisation of the cells in that manuscript. This is important because the large majority of results were generated using OV2008, C13* and SKOV3 cells.

Editorial

Language editing. Some of the language used requires editing. For example

1. In the abstract, the expression 'a bunch of transcription factors' is used – this is too colloquial
2. The expression "CEBPB expression dramatically affected patient outcomes (Fig. 2c and Supplementary Fig. S4b)." on lines 138 – 9. This is not true. The data show that CEBPB expression was significantly associated with altered patient outcomes, which is not the same thing.
3. Line 161 – 2. Patients are not routinely treated with cisplatin – nearly all patients receive carboplatin.

Reviewer #3 (Remarks to the Author):

The manuscript entitled "C/EBP β 1 enhances chemoresistance of tumor cells by reprogramming H3K79 methylation" described epigenomic reprogramming associated with acquired chemoresistance in ovarian cancer and pinpointed global alterations in histone methylation. Intriguingly but interestingly, those epigenomic events seem to be associated with an upstream regulator, C/EBP β . The manuscript is well-written and the illustrations were nicely done; the ChIP-seq data appear to be convincing. The immunohistochemistry results showing nuclear positivity of C/EBP β in a subset of tumors cells among the bulk recurrent ovarian tumor cells (Figure 3g) may further support the enrichment or reactivation of C/EBP β transcription during development of chemoresistance. The potential interaction between C/EBP β and its nuclear co-factor, TODIL (a histone methyl transferase), and their molecular collaboration in modulating histone methylation and gene transcription are well-described and the results appear to be solid and cogent. Nevertheless, there are a few questions which remain to be addressed:

Major:

1. The rationale of selecting C/EBP β as the major transcription hub in chemoresistant ovarian cancer remains relatively weak.

It would be more convincing if several key findings such as overexpression of C/EBP β and clinical prognosis value of C/EBP β can be validated using an independent dataset of ovarian tumors. The authors are encouraged to look into the RNA-seq dataset published by DD Bowtell's group Nature (PMID: 26017449) which include from primary tumors and post-treatment ovarian tumors to further confirm their findings.

2. Small molecular inhibitor, SGC0946, was used to block TODIL activity; however, the specificity of SGC0946 was not described or demonstrated in this manuscript. The authors should perform additional experiments to address this issue. For example, the authors may need to demonstrate inhibition of TODIL target genes by SGC0946 at a dose-dependent manner. Other inhibitor(s) of TODIL should be used to confirm that the observed phenotypes are not off-target effects

Minor:

In addition to reference 17, another most recent review may also be cited: PMID: 27012190.

Reviewer #1

1. The Introduction is very difficult to read and should be rewritten to carefully pose the question that will be addressed in the manuscript.

Answer:

According to the reviewer's suggestion, we have carefully rewritten the introduction to pose the question that will be addressed in the manuscript. We did our best to improve the introduction to be easier for reading. Additionally, English language editing of the revised manuscript has been done by Elsevier Language Editing Services before resubmission.

2. Page 5. While a fraction of high-grade serous are now thought to arise from the fallopian tubes, there are other sites that likely serve as the site of origin. The language at the beginning of this page around this topic should be carefully checked.

Answer:

Indeed, high-grade serous ovarian may also arise from the ovarian surface epithelium except for the fallopian tube. According to the reviewer's suggestion, we replaced the original sentence with another sentence: "Increasing evidence indicates that a significant part of HG-SOC originates in the fimbriae of fallopian tube" (page 5, line 90–91).

We appreciate the reviewer for the suggestion.

3. Page 5. For epithelial cell isolations from the fallopian tubes, what kind of "benign diseases" did they have? What portion of the fallopian tube was used? Were the two groups age matched? This is important since there could be age-specific differences in methylation profiles.

Answer:

The brief information for patients with benign diseases was described in the original Supplementary Methods: 'benign diseases' included 10 patients with hysteromyoma and 10 patients with adenomyosis underwent hysterectomy with bilateral salpingectomy. The HE staining of fallopian tubes were reviewed by two pathologists to confirm no significant pathologic changes.

We used the fimbriae of the fallopian tube, because this portion of the fallopian tube is considered most likely to be the origin of HG-SOC.

There was no statistical difference in age between the two groups ($P = 0.161$).

In response to the reviewer's concern, these descriptions were added to Results section (page 5, line 91–93) and the Methods section (page 24, line 506–511) of the revised manuscript.

4. Figure 1b and page 5. It is unclear why borderline ovarian tumors were used. Borderline tumors are thought to be more of a precursor for low-grade serous cancers, but not for high-grade serous tumors. Inclusion of borderline tumors would seem irrelevant for the present study.

Answer:

We originally intended to provide more information about H3K79 methylation levels in various

histologic types of ovarian tumors. In response to the reviewer's concern, we removed the data of borderline tumors from the manuscript and figures, which indeed does not influence the main conclusion in the manuscript.

We appreciate the reviewer for indicating this issue.

5. Supplementary table 2. Contemporary cut-off for residual disease would be R0 (i.e., no gross residual) or R1 (any residual). If such data are available, then authors may want to consider presenting that.

Answer:

According to the reviewer's suggestion, we re-performed the Cox regression analysis using 'R0 v R1' as the cut-off values for residual disease. The data were added to Results section (page 8, line 165–171), Table 1, Supplementary Table S2 and S3 of the revised manuscript.

6. Page 5 and figure 1d. The data are difficult to follow. What was the apparent strength of correlation? Such correlation values should be presented. It is important to understand such data carefully since some correlation may be statistically significant but functionally irrelevant. Also, on the figure, some of the p-values are cut off from the figure.

Answer:

The related results might not be clearly described in the original manuscript due to the oversimplification and the layout of the related data was inappropriate. The original Figure 1d and 1e were used to show the groundwork for the potential correlation between C/EBPβ-associated H3K79me2/me3 changes and C/EBPβ-associated H3K9me3 changes, which were further analyzed in original Figure 4 and Supplementary Figure 10. These related results might be difficult for understanding because they were far apart from each other and also because few words were used to explain this finding.

Indeed, some correlation may be statistically significant but functionally irrelevant. The correlation in the original Figure 1d and 1e led us to perform further investigation (in original Figure 4 and Supplementary Figure 10). However, the results turn out to be negative.

According to the reviewer's suggestion, we combined the original Figure 1d and 1e to the original Supplementary Figure 10 (as new Supplementary Figure 13) and discussed concerns more intensively. The related description was added to Results section (page 12, line 244–259) corresponding to new Supplementary Figure 13 of the revised manuscript.

7. Figure 2a and page 6. What was the level of stringency for the motif analysis?

Answer:

The level of stringency for the motif analysis was $P < 1 \times 10^{-11}$; the P -value for CEBPB motif was 1×10^{-86} . Although we plotted the ' $-\log_2 P$ value' in original Figure 2a (new Figure 1d), the data were not intuitive. In response to the reviewer's concern, the related description was added to Results section (page 6, line 124–126).

8. Figure 2. Again, it is unclear why borderline tumors are included in this analysis.

Answer:

We originally intended to provide more information about C/EBP β expression in various histologic types of ovarian tumors. In response to the reviewer's concern, we removed the data of borderline tumors from the manuscript and figures, which indeed does not influence the main conclusion in the manuscript.

We appreciate the reviewer for indicating this issue.

9. Figure 2b. How were the cut-off levels selected for “high” versus “low”?

Answer:

The cut-off levels selected for “high” versus “low” was described in the Figure legend of Figure 2b: “Prognosis analysis using TCGA dataset. Samples were divided into low- and high-expression groups based on the median of each gene”.

In response to the reviewer's concern, we added the related description to the Results section of the revised manuscript (page 7, line 145–147).

We appreciate the reviewer for indicating this issue.

10. Figure 2f. As above, what were “high” versus “low” levels?

Answer:

The definition of “high” versus “low” levels was described in the Methods section: “HSCORE < 1.5 was classified as a low protein level, and HSCORE \geq 1.5 was classified as a high protein level”. The description of the immunoreactivity-scoring system (HSCORE, range from 0 to 3) was also provided in the Methods section. And representative images of different C/EBP β staining intensity were given in original Supplementary Figure S5 (new Supplementary Figure S7).

To make it more convenient for reading, we added the related description to the Results section of the revised manuscript (page 8, line 159–160).

We appreciate the reviewer for indicating this issue.

11. Page 7. Would suggest avoiding vague terms such as “dramatic” with regard to patient outcomes. This is simply a correlational study and language suggesting causality should be avoided.

Answer:

According to the reviewer's suggestion, we deleted this sentence. We have carefully revised the manuscript to avoid this kind of problem.

12. Page 9. Fairly limited analyses are presented for other chemotherapy drugs. It would be important to understand whether the potential effects are indeed restricted to platinum or if other commonly used chemotherapy drugs (e.g., doxorubicin, topotecan) are also affected.

Answer:

According to the reviewer's suggestion, we investigated the effect of C/EBP β on the sensitivity of other commonly used chemotherapy drugs. We found that knockdown of C/EBP β increased sensitivity to carboplatin, which has an antitumor mechanism similar to cisplatin. Our preliminary data showed that C/EBP β levels have no statistically significant impact on paclitaxel, doxorubicin, and topotecan sensitivity. However, much work is needed to further determine the role of C/EBP β on the sensitivity of these drugs,

the investigation of which is beyond the scope of the present study. The data was added to Results section (page 10, line 210–215) and new Supplementary Figure S11 of the revised manuscript. We also revised the title of manuscript ‘C/EBPβ enhances chemoresistance...’ to ‘C/EBPβ enhances platinum resistance...’ to address our topic more precisely (page 1, line 1).

13. Figure 3h. While some changes in CEBPB may be seen in patient samples after exposure to platinum, what would be important to understand is whether there were differences in PFS based on CEBPB level changes.

Answer:

The results herein suggested that C/EBPβ level changes after exposure to platinum might be due to post-transcriptional mechanisms, while mRNA levels were not significantly altered (new supplementary figure S12). The data were added to the results section of the revised manuscript (page 10, line 216 to page 11, line 232). Although there are studies investigated recurrent disease on the level of mRNA which used ascites specimens (Patch, A, et al. Nature 2015; 521(7553): 489-494), investigation of C/EBPβ requests quantification of protein levels, such as immunohistochemical (IHC) analysis using tissue specimens. However, operations for recurrent ovarian cancer were less common, and tissue specimens obtained in both initial surgery and re-operation on recurrent disease of the same patient were rare. Therefore, the sample size was too small to analyze the PFS and/or chemotherapeutic response after treatment of recurrent tumor.

‘Progress-free survival’ is typically referred to initial treatment. IHC analysis in human HG-SOC specimens showed that the PFS in C/EBPβ protein high group was significantly shorter than that in C/EBPβ protein low group. The result was shown in original Figure 2f (new Figure 2e).

14. Figure 4d. Most of the gene changes appear to be relatively modest. Were there any (if so, how many) that exceeded the typical 2-fold cut-off used? What was the extent of knock-down seen for CEBPB with the siRNA?

Answer:

The results in Figure 4d may be not nice, but still reliable. There are 1114 genes in C13* cells and 780 genes in SKOV3 cells that showed the expression changes exceeding 2-fold cut-off. The extent of knock-down in C/EBPβ protein levels were 5.97 fold for C13* cells and 6.71 fold for SKOV3 cells, shown in Figure 3b and Supplementary Figure 7a. We chose the genes that met all of the three criteria: (1) consistently and differentially expressed in two cell lines, (2) directly targeted by C/EBPβ in ChIP-seq, (3) showed decreased H3K79 methylation in C/EBPβ knocked down cells. The inclusion criteria were very strict, thus raising the cut-off value may loss many useful information. In RT-qPCR verification, all of the selected genes exceed 1.5-fold change and many of them exceed 2-fold change (Figure 5h and new Supplementary Fig. S16c). Some of the representative genes were also verified using Western blotting (Figure 5i and new Supplementary Figure S19b).

15. Page 13. While the authors acknowledge various mechanistic possibilities, it would be more convincing if definitive mechanistic insights were gained/provided.

Answer:

This section of the present study intended to investigate only one mechanism: whether C/EBP β -mediated cisplatin resistance is depended on the effect of C/EBP β in regulating H3K79 methylation, in other words, whether depended on H3K79 methyltransferase DOT1L. Therefore, we looked into the downstream effects of C/EBP β -DOT1L co-targeted genes in order to analyze whether C/EBP β played a role in these effects as well as whether these effects of C/EBP β were mediated by DOT1L. Indeed, the results confirmed this hypothesis. However, considering the large number of downstream genes and the complex network, more researches are needed to understand the precise role of these genes and maybe more other genes in C/EBP β -induced cisplatin resistance. We will investigate that in future in our ongoing projects. We rewrote the Results section related to Figure 6 to better clarify the mechanism that intended to be addressed in the manuscript (page 15, line 316 to page 17, line 351).

16. There are multiple grammatical problems that should be carefully fixed.

Answer:

We have carefully revised the manuscript to avoid grammatical problems. English language editing of the revised manuscript has been done by Elsevier Language Editing Services before resubmission.

Reviewer #2:

In this manuscript, the authors present data to support a role for C/EBP β as a driver of platinum resistance and poor outcome in high grade serous ovarian cancer.

I found the manuscript dense and rather difficult to follow. However, the hypothesis is that expression of C/EBP β (via transcriptional upregulation of CEBPB) in HGSOC induces altered gene expression via an interaction with the methyltransferase DOT1L and subsequent H3K79 methylation. The altered gene expression results in more rapid cell growth and resistance to platinum – multiple potential genes are involved in the platinum resistance. In patient samples and cohorts, increased transcription of CEBPB and expression of C/EBP β are both associated with poorer outcome and increased likelihood of relapsing in a platinum-resistant timeframe.

The manuscript presents a very amount of information. A large number of techniques have been employed across many samples, and overall this is technically impressive work. It should be noted that the first description of increased C/EBP β expression in ovarian cancer came in 1999 (Sundfelt et al Br. J. Cancer 79:1240), so this is not entirely novel. However, the data suggesting increased expression and altered histone methylation are largely convincing.

Major points

The main issue is the big picture. Currently, this manuscript is a very detailed description of biology of C/EBP β . However, the manuscript fails to address the overall landscape of ovarian/fallopian high grade serous carcinoma. This is a disease of universal TP53 mutation and extreme copy number alteration. The large consortia, in particular TCGA and ICGC, have revealed a great deal about the genomic drivers, gene expression classifiers and copy number states of these tumours. There are, in addition, many known

important biological factors in this disease (e.g. germline BRCA1/2 mutation, BRCA1 promoter methylation, CCNE1 amplification, PTEN loss etc etc). Having read this manuscript several times, I still do not understand two vital pieces of information: What is the relationship of C/EBP β expression to the myriad of other important factors in HGSC, and what drives CEBPB upregulation?

Answer:

According to the reviewer's suggestion, the first part of result section was revised to better address the overall landscape of HG-SOC. The differentially expressed genes in HG-SOC showed a high enrichment in TP53 associated signaling pathways. The results were added as new Supplementary Figure S2 (page 5, line 100–102). We investigated protein interactions among epigenetically altered genes in HG-SOC and known important biological factors in this disease. Genes with increased H3K4 methylation showed high enrichment in those groups of genes that interact with MYC, TP53, BRCA1, RB1, and CCNE1. The results were added as new Supplementary Figure S5 (page 7, line 134–141). The Motif analysis might be the most important result on the overall landscape of HG-SOC. This result showed a novel view of the regularity in this highly complicated disease. We removed the results from original Figure 2A to new Figure 1d. The related description was also revised (page 6, line 121–134).

C/EBP β expression might play an important role in BRCA1 related DNA damage response, as shown in original Supplementary Figure S11 (new Supplementary Figure S14) (page 13, line 267–271). Although C/EBP β did not regulate BRCA1 expression, knock-down of C/EBP β led to the decreased expression of many genes, such as BRCA2, ATM, EGFR, etc., in BRCA1-related DNA damage response pathway. Ingenuity pathway analysis (IPA) further predicted the suppression of BRCA1-related DNA damage repair in C/EBP β knock-down cells. Our research proposed many probable mechanisms in C/EBP β /DOT1L-mediated chemoresistance. In this study, we mainly focused on investigating whether and how C/EBP β regulated H3K79 methylation and focused on determining whether C/EBP β -mediated chemoresistance was depended on the ability of C/EBP β in regulating H3K79 methylation. There might be various downstream mechanisms of C/EBP β -DOT1L in promoting chemoresistance, BRCA1-related DNA damage response might being one of them. Thanks for the reviewer's advices, investigating the relationship of C/EBP β expression to the other important factors (e.g. BRCA1/2, CCNE1, etc.) in HG-SOC could provide important insights into this disease. Since the topic of this study is 'C/EBP β mediated epigenetic reprogramming', we will investigate the relationship of C/EBP β and BRCA1/2 in future in our ongoing projects. In discussion section, we added a paragraph to discuss the research prospects of C/EBP β in cancer (page 19, line 411 to page 20, line 421).

Another question was what drives C/EBP β upregulation. It was reported that post-transcriptional regulation was a key mechanism for the regulation of C/EBP β protein, and that mRNA levels would not necessarily be regulated (Zahnow CA. Expert reviews in molecular medicine 2009, 11: e12). Consistent with this, while C/EBP β protein was remarkably higher in C13* cells as compare to its parent OV2008 cells, there is no difference in the CEBPB mRNA levels between the two cell lines (new Supplementary Fig. S12a). Similarly, there are no significant changes in CEBPB mRNA after cisplatin treatment *in vitro* (new Supplementary Fig. S12b) or *in vivo* (new Supplementary Fig. S12c). These results suggested that C/EBP β changes after exposure to platinum might be primarily due to post-transcriptional mechanisms. However, CEBPB mRNA was significantly increased in ovarian cancer (new Supplementary Fig. S12e), and the mRNA versus protein levels of C/EBP β was positively correlated among cell lines (new Supplementary Fig. S12a) and clinical specimens (new Supplementary Fig. S12f), indicating the involvement of transcriptional

regulation of C/EBP β . The data were added to the results section of the revised manuscript (page 10, line 216 to page 11, line 232). However, there are numerous probable mechanisms for the transcriptional regulation of C/EBP β as we discussed in the revised manuscript (page 21, line 437–448). Ovarian cancer is a complicated tumor with no clear etiological factor. Therefore, a lot of works are needed to find out a convincing mechanism for C/EBP β upregulation in ovarian cancer. Additionally, our findings suggested that there may be different mechanisms for C/EBP β upregulation in tumorigenesis and C/EBP β upregulation after chemotherapy. In this study, we mainly focused on C/EBP β functions and the underlying mechanisms. We will investigate what driving CEBPB upregulation in future in our ongoing projects.

Minor points

There are a series of minor technical issues that need to be addressed:

1. Identification of primary cells. The first figure in the manuscript relates to differential methylation in primary HGSOE cells vs normal “oviduct” (the term ‘fallopian tube’ is more widely used). However, it is difficult from the supplementary methods to understand how the fallopian tube cells were isolated and purified. Moreover, it states that primary ascites culture in MCDB/M199 medium was used to purify tumour cells – the only method for identification of these as tumour cells was EpCAM staining. However, EpCAM staining alone is insufficient. There needs to be confirmation of lineage (e.g. PAX8) and confirmation of mutant TP53 in the cultured tumour cells.

Answer:

According to the reviewer’s suggestion, the term “oviduct” was replaced by “fallopian tube”. In addition to EpCAM staining, the lineage was confirmed by PAX8 staining and the mutant of TP53 in each HG-SOC specimens was confirmed by whole exon sequencing. These data were added to Results section (page 5, line 94–102), the revised Supplementary Methods, Supplementary Fig. S1 and Supplementary Tables S1.

2. Clinical data. In 2005, FIGO changed the grading of serous carcinomas into a binary grading – low grade serous carcinoma and high grade serous carcinoma. The clinical data in supplementary table 1 state that some cases were grade 2. These tumours were thus either collected prior to 2005 or graded according to the old system. Pathology review should be performed.

Answer:

Pathology review had already been performed by pathologists before the initial submission of the manuscript, and serous carcinomas were graded according the two-tier grading system. We added this statement to Methods section (page 24, line 516–517) of the revised manuscript. We originally intended to provide the information of histologic grade according to the old system. However, this seems to be meaningless and makes readers confused. According to the reviewer’s suggestion, this information was replaced by the statements “Histologic type” and “HG-SOC” in the revised Supplementary Table 1.

3. Similarly, the univariate and Cox proportional hazards model uses grade as one of the analyses. However, all high grade serous carcinomas are, by definition, high grade, and so it is not possible to undertake an analysis by grade.

Answer:

According to the reviewer's suggestion, we remove the data of histologic grade from the Manuscript and Tables.

4. Furthermore, there are two other features of concern in these univariate/multivariate analyses. The first is age. Increased age is widely accepted to be a poor prognostic factor (as revealed in multiple analyses), but was not significant here. This may relate to the age cut-off of 50 years. Why was 50 used as the cut-off? The median age of diagnosis of ovarian cancer is over 60 (in this reviewer's centre, it is 65.2 years) and HGSOE cases aged <50 are rare (in this reviewer's centre, only 6.2% HGSOE patients were aged <50 at time of diagnosis). The second is stage. Stage is a critical prognostic factor but was again negative here. This may relate to the classifiers used (stage II and III vs IV). More appropriate is stage II vs stages III and IV, given the clinical and prognostic significance of dissemination beyond the pelvis in stage III. Overall, these analyses need to be re-performed.

Answer:

According to the reviewer's suggestion, we re-performed the univariate/multivariate analyses. Data were added to Results section (page 8, line 165–171), the revised Table 1 and Supplementary Tables 3–4. It was reported that the overall incidence rate of ovary cancer in our country significantly increased from age 40-50 years, and the incidence rate among patients aged < 55 years old was approximately 30% (Wei K, et al. Chin J Cancer Res 2015, 27(1): 38-43). However, more specimens might be obtained from younger patients. It might be because of that younger patients are more willing to receive operation, while older patients are more willing to receive conservative non-surgical treatment. Therefore, the median age of patients for obtaining tissue specimens was lower than the overall morbidity age. Thus 55 may be a more proper cut-off of age in the present study.

If we used all of our epithelial ovarian cancer specimens (including all FIGO stages), age was shown to be risk factor for OS (HR 2.920, 95% CI 1.466–5.815, $P = 0.002$) and PFS (HR 2.403, 95% CI 1.302–4.435, $P = 0.005$) of patients in multivariate analyses. Interestingly, when restricted to advanced-stage (stage II–IV) HG-SOC, age and FIGO stage were not significant risk factors in multivariate analyses of the present study. Age (≤ 55 v > 55 years) and FIGO stage (II v III, IV) were shown only to be risk factors for overall survival of patients with HG-SOC in univariate analysis, barely reached statistical significance (new Supplementary Table 3). These results may also be due to limited sample size. Nevertheless, these do not influence the main conclusion of this study, as we mainly intended to determine whether C/EBP β was a risk factor in ovarian cancer and the known major factors (age, stage, ascites and amount of residual disease) showed no bias between C/EBP β protein low- and high- groups (Supplementary Table S2).

5. Justification is required for the method to delineate tumours as 'CEBPB high' vs 'CEBPB low', as well as 'C/EBP β high' vs 'C/EBP β low' in Figure 2. Why were the stated cut-off values used?

Answer:

In original Figure 2b–2c (new Figure 2a–2c), dozens of the genes were individually subjected to prognosis analysis using public dataset in batches. To improve comparability of the results, 'high' versus 'low' levels of the expression of each gene was defined using a uniform cut-off threshold, the median expression of each gene, which is generally accepted as a commonly used threshold. The definition of

“high” versus “low” levels was described in the Figure Legend of original Figure 2b. To make it more convenient for reading, we added the related description to the Results section of the revised manuscript (page 7, line 146–147).

In original Figure 2f (new Figure 2e, 2f), prognosis analysis of C/EBP β protein was performed based on immunohistochemical analysis using an immunoreactivity-scoring system (HSCORE, range from 0 to 3). Thus, it is also generally accepted to define ‘high’ versus ‘low’ protein levels using the mean value HSCORE 1.5. The definition of “high” versus “low” levels was described in the original Methods section: ‘HSCORE < 1.5 was classified as a low protein level, and HSCORE \geq 1.5 was classified as a high protein level’. The description of the immunoreactivity-scoring system (HSCORE, range from 0 to 3) was also provided in the Methods section. And representative images of different C/EBP β staining intensity were given in original Supplementary Figure S5 (new Supplementary Figure S7). To make it more convenient for reading, we added the related description to the Results section of the revised manuscript (page 8, line 159–160).

6. Cell lines. The authors will be aware that the provenance of many ‘high grade serous’ ovarian cancer cell lines has recently been questioned (Domcke et al Nature Commun. 2013). Of the cell lines used by the authors, only CAOV3 is thought to be ‘likely high grade serous’ by Domcke et al. ES-2 and OV90 are ‘possibly high grade serous’ and SKOV3 is listed as ‘unlikely’. I cannot find details of OV2008 on either the COSMIC or CCLE databases. Reference 44 is cited to describe both OV2008 and C13*, but there is no description of the generation or characterisation of the cells in that manuscript. This is important because the large majority of results were generated using OV2008, C13* and SKOV3 cells.

Answer:

C/EBP β protein levels were tested in a series of ovarian cancer cell lines. The cisplatin-sensitive OV2008 cells were originally established from a patient with serous cystadenocarcinoma of the ovary, and the cisplatin-resistant C13* cells were generated from OV2008 cells by monthly *in vitro* selection with 1 μ M cisplatin (Andrews PA. Cancer Res 1985, 45(12 Pt 1): 6250-6253). OV2008 and C13* were not yet been classified using technology similar to that used by Domcke et al. Nevertheless, the cisplatin-sensitive and -resistant cell lines pair OV2008 and C13* were good model for investigating chemoresistance in the present study. That is because C/EBP β was negligibly expressed in cisplatin-sensitive OV2008 and was strongly expressed in cisplatin -resistant C13*, and because they represented the shift in C/EBP β expression upon cisplatin treatment.

According to the reviewer’s suggestion, we added the description of the generation of the cells to the Methods section of the revised manuscript (page 22, line 461–465). In addition, we added data preformed using another HG-SOC cell line Caov4 to further confirm the results of this manuscript. The results showed that C/EBP β promoted cisplatin resistance in Caov4 cells, and C/EBP β also regulated H3K79 methylation to promote gene expression in Caov4 cells. These data were added to Results section (page 9, line 180; page 14, line 290), the revised Supplementary Fig. S8, S9 and S16d.

Editorial

Language editing. Some of the language used requires editing. For example

1. In the abstract, the expression ‘a bunch of transcription factors’ is used – this is too colloquial
2. The expression “CEBPB expression dramatically affected patient outcomes (Fig. 2c and Supplementary Fig. S4b).” on lines 138 – 9. This is not true. The data show that CEBPB expression was significantly

associated with altered patient outcomes, which is not the same thing.

3. Line 161 – 2. Patients are not routinely treated with cisplatin – nearly all patients receive carboplatin.

Answer:

According to the reviewer's suggestion, the sentence 'a bunch of transcription factors' was replaced by 'a series of' (page 2, line 28); the sentence 'CEBPB expression dramatically affected patient outcomes' was deleted. When referred to clinical data, 'cisplatin' was replaced by 'platinum' (page 9, line 176). We appreciate the reviewer for indicating the inappropriate statements, and we have carefully revised the language of the manuscript to avoid this kind of problem. Before resubmission, English language editing of the revised manuscript has been done by Elsevier Language Editing Services.

Reviewer #3 (Remarks to the Author):

The manuscript entitled "C/EBP β 1 enhances chemoresistance of tumor cells by reprogramming H3K79 methylation" described epigenomic reprogramming associated with acquired chemoresistance in ovarian cancer and pinpointed global alterations in histone methylation. Intriguingly but interestingly, those epigenomic events seem to be associated with an upstream regulator, C/EBP β . The manuscript is well-written and the illustrations were nicely done; the ChIP-seq data appear to be convincing. The immunohistochemistry results showing nuclear positivity of C/EBP β in a subset of tumors cells among the bulk recurrent ovarian tumor cells (Figure 3g) may further support the enrichment or reactivation of C/EBP β transcription during development of chemoresistance. The potential interaction between C/EBP β and its nuclear co-factor, TODIL (a histone methyl transferase), and their molecular collaboration in modulating histone methylation and gene transcription are well-described and the results appear to be solid and cogent. Nevertheless, there are a few questions which remain to be addressed:

Major:

1. The rationale of selecting C/EBP β as the major transcription hub in chemoresistant ovarian cancer remains relatively weak.

It would be more convincing if several key findings such as overexpression of C/EBP β and clinical prognosis value of C/EBP β can be validated using an independent dataset of ovarian tumors. The authors are encouraged to look into the RNA-seq dataset published by DD Bowtell's group Nature (PMID: 26017449) which include from primary tumors and post-treatment ovarian tumors to further confirm their findings.

Answer:

According to the reviewer's suggestion, we analyzed the RNA-seq dataset published by DD Bowtell's group Nature (PMID: 26017449). *CEBPB* mRNA levels were showed to be slightly but not significantly elevated in recurrent diseases compared with primary tumors (new Supplementary Fig. S12d). It was reported that post-transcriptional regulation was a key mechanism for the regulation of C/EBP β protein, and that mRNA levels would not necessarily be regulated (Zahnow CA. Expert reviews in molecular medicine 2009, 11: e12). Consistent with this, although we found that the level of C/EBP β protein was remarkably higher in C13* cells compared to that found in its parent OV2008 cells, there was no difference in *CEBPB* mRNA levels between the two cell lines (Supplementary Fig. S12a). Similarly, there were no significant changes in *CEBPB* mRNA levels after cisplatin treatment *in vitro* (Supplementary Fig. S12b)

and *in vivo* (Supplementary Fig. S12c). These results suggest that changes to C/EBP β after platinum exposure may be primarily because of post-transcriptional mechanisms. However, *CEBPB* mRNA was significantly increased in ovarian cancer (Supplementary Fig. S12e), and there was a positive correlation between C/EBP β mRNA and protein levels among cell lines (Supplementary Fig. S12a) and clinical specimens (Supplementary Fig. S12f), indicating the involvement of transcriptional regulation of C/EBP β . The data were added to the results section of the revised manuscript (page 10, line 216 to page 11, line 232). Therefore, our findings suggested that there may be different mechanisms for C/EBP β upregulation in tumorigenesis and after chemotherapy, the investigation of which is beyond the scope of the present study. In this study, we mainly focused on C/EBP β functions and the underlying mechanisms. We will investigate what driving CEBPB upregulation in future in our ongoing projects.

2. Small molecular inhibitor, SGC0946, was used to block TODIL activity; however, the specificity of SGC0946 was not described or demonstrated in this manuscript. The authors should perform additional experiments to address this issue. For example, the authors may need to demonstrate inhibition of TODIL target genes by SGC0946 at a dose-dependent manner. Other inhibitor(s) of TODIL should be used to confirm that the observed phenotypes are not off-target effects

Answer:

According to the reviewer's suggestion, we added the data from the experiments using another DOT1L inhibitor EPZ004777 to the revised manuscript. DOT1L targeted-genes *MEIS1* and *NANOG* were analyzed and the results showed that SGC0946 and EPZ004777 inhibit the expression of *MEIS1* and *NANOG* at a dose-dependent manner, confirming the inhibition of DOT1L activity in ovarian cancer cells. The related description was added to Results section (page 15–16) and Figure legends corresponding to new Figure 6, new Supplementary Fig. S20, S23 and S24.

Minor:

In addition to reference 17, another most recent review may also be cited: PMID: 27012190.

Answer:

According to the reviewer's suggestion, the reference (PMID: 27012190) was cited (page 5, line 91; page 30, line 654–655).

With best regards.

REVIEWERS' COMMENTS:

Reviewer #1 (Remarks to the Author):

My comments have been addressed

Reviewer #2 (Remarks to the Author):

In this revised manuscript, the authors have addressed a wide range of comments from all three reviewers.

With reference to my specific comments, the writing has been significantly improved – although the manuscript remains dense and contains a large amount of data, the flow is more logical and it is easier to follow. More specifically the association between C/EBP β and characteristics of ovarian high grade serous carcinoma are now clearer.

The mechanisms of C/EBP β upregulation, however, remains uncertain – there are intriguing discussions about potential mechanisms, some of which are transcriptional and some post-transcriptional. There is an association between C/EBP β and resistance to platinum (cisplatin and carboplatin) but not other chemotherapy drugs (paclitaxel, doxorubicin and topotecan). My original comments about the appropriateness of the cell lines remain, but the authors have undertaken a wide series of experiments to understand the role of C/EBP β .

Overall, the authors present a plausible putative pathway whereby C/EBP β and DOT1L together increase methylation of H3K79, which thus upregulates expression of oncogenic genes and drives poor platinum response and poor survival.

Minor points

Although the writing has been improved greatly, there remain some minor errors:

Line 48: 'that' should be replaced by 'those' or potentially even 'mechanisms'

Line 90: 'part' should be replaced by 'proportion'.

Line 100: 'that' should be replaced by 'those'.

Line 223: 'showed' should be replaced by 'shown'.

Line 437: 'upregulates' should be replaced by 'is upregulated'.

Reviewer #3 (Remarks to the Author):

The clarity of presentation has been improved in the revised manuscript. The authors have adequately addressed most of the concerns.

i) Authors' response to point 6 of Reviewer #1.

The new Supplemental Figure 13 demonstrating that shCEBPB affects regional and global changes in H3K79me2/me3. The conclusion derived from this part of data seems to be justified.

However, the reviewer found a technical limitation associated with ChIP-Seq related experiments. Please see Encode guidelines on ChIP-seq experimental design and practices for details (Genome Res. 2012 Sep; 22(9): 1813–1831). If the ChIP-Seq was not performed in triplicate, the authors should address this concern in Discussion and should interpret their ChIP-seq data in a careful way. Alternatively, the ChIP binding events can be validated by another antibody specific to C/EBP β . The authors should at least point out this pitfall.

ii) As pointed out in point 6 by reviewer #2, SKOV3 is unlikely to be an "ovarian" serous tumor cell line (Domcke et al Nature Commun. 2013). Therefore, there is a question of generalizing current findings.

The authors could indicate that since SKOV3 harbors mutations in TP53, PIK3CA, ARID1A, and gene amplification in ERBB2 and CCNE1 loci, which are often identified in high-grade "ovarian" endometrioid tumor or in the "uterine" serous tumor, there is a likelihood that SKOV3 initially derived from either one of these histological types rather than from the "ovarian" serous type. To support this view, the authors could cite the following references reporting hallmark genetic markers in the above histological types of GYN cancers (Domcke et al Nature Commun. 2013 PMID: 23839242; PMID: 26321251; PMID: 22923510; PMID: 22888282 , and COSMIC cell line mutation data.)

Reviewer #1

My comments have been addressed

Reviewer #2:

In this revised manuscript, the authors have addressed a wide range of comments from all three reviewers.

With reference to my specific comments, the writing has been significantly improved – although the manuscript remains dense and contains a large amount of data, the flow is more logical and it is easier to follow. More specifically the association between C/EBP β and characteristics of ovarian high grade serous carcinoma are now clearer.

The mechanisms of C/EBP β upregulation, however, remains uncertain – there are intriguing discussions about potential mechanisms, some of which are transcriptional and some post-transcriptional. There is an association between C/EBP β and resistance to platinum (cisplatin and carboplatin) but not other chemotherapy drugs (paclitaxel, doxorubicin and topotecan). My original comments about the appropriateness of the cell lines remain, but the authors have undertaken a wide series of experiments to understand the role of C/EBP β .

Overall, the authors present a plausible putative pathway whereby C/EBP β and DOT1L together increase methylation of H3K79, which thus upregulates expression of oncogenic genes and drives poor platinum response and poor survival.

Answer:

In response to the reviewer's concern, we added a paragraph to the discussion section addressing limitations to this study, in regard to the appropriateness of the cell lines (page 22, line 474 – page 23, line 487).

Minor points

Although the writing has been improved greatly, there remain some minor errors:

Line 48: 'that' should be replaced by 'those' or potentially even 'mechanisms'

Line 90: 'part' should be replaced by 'proportion'.

Line 100: 'that' should be replaced by 'those'.

Line 223: 'showed' should be replaced by 'shown'.

Line 437: 'upregulates' should be replaced by 'is upregulated'.

Answer:

According to the reviewer's suggestion, the above errors were corrected.

Reviewer #3 (Remarks to the Author):

The clarity of presentation has been improved in the revised manuscript. The authors have adequately addressed most of the concerns.

i) Authors' response to point 6 of Reviewer #1.

The new Supplemental Figure 13 demonstrating that shCEBPB affects regional and global changes in H3K79me2/me3. The conclusion derived from this part of data seems to be justified.

However, the reviewer found a technical limitation associated with ChIP-Seq related experiments. Please see Encode guidelines on ChIP-seq experimental design and practices for details (Genome Res. 2012 Sep; 22(9): 1813–1831). If the ChIP-Seq was not performed in triplicate, the authors should address this concern in Discussion and should interpret their ChIP-seq data in a careful way. Alternatively, the ChIP binding events can be validated by another antibody specific to C/EBP β . The authors should at least point out this pitfall.

ii) As pointed out in point 6 by reviewer #2, SKOV3 is unlikely to be an “ovarian” serous tumor cell line (Domcke et al Nature Commun. 2013). Therefore, there is a question of generalizing current findings.

The authors could indicate that since SKOV3 harbors mutations in TP53, PIK3CA, ARID1A, and gene amplification in ERBB2 and CCNE1 loci, which are often identified in high-grade “ovarian” endometrioid tumor or in the “uterine” serous tumor, there is a likelihood that SKOV3 initially derived from either one of these histological types rather than from the “ovarian” serous type. To support this view, the authors could cite the following references reporting hallmark genetic markers in the above histological types of GYN cancers (Domcke et al Nature Commun. 2013 PMID: 23839242; PMID: 26321251; PMID: 22923510; PMID: 22888282 , and COSMIC cell line mutation data.)

Answer:

According to the reviewer's suggestion, we added a paragraph to the discussion section addressing limitations to this study, in regard to the appropriateness of the cell lines and the technical limitation associated with ChIP-Seq (page 22, line 474 – page 23, line 487).

With best regards.

Qing-Lei Gao

Cancer Biology Research Center (Key laboratory of the ministry of education)

Tongji Hospital, Tongji Medical College

Huazhong University of Science and Technology

1095 Jiefang Anv. Wuhan, 430030, China

Tel: 86-27-83663351; Fax: 86-27-83662681;

E-mail: qingleigao@hotmail.com